# Quantification and optimization of platinum–molybdenum carbide interfacial sites to enhance low-temperature water-gas shift reaction

Ruiying Li[1], Jingyuan Shang[1], Fei Wang[1], Qing Lu[1], Hao Yan [1], Yongxiao Tuo[1], Yibin Liu [1]✉, Xiang Feng [1]✉, Xiaobo Chen[1], De Chen[2]✉ & Chaohe Yang[1]✉

Pt/α-MoC$_{1-x}$ catalysts exhibit exceptional activity in low-temperature water-gas shift reactions. However, quantitatively identifying and fine-tuning the active sites has remained a significant challenge. In this study, we reveal that fully exposed monolayer Pt nanoclusters on molybdenum carbides demonstrate mass activity that exceeds that of bulk molybdenum carbide catalysts by one to two orders of magnitude at 100–200 °C for low-temperature water-gas shift reactions. This advancement is driven by the precise quantification and elucidation of active sites along the Pt-molybdenum carbide interfacial perimeter. By combining sacrificial CO adsorption per Pt atom, Density Functional Theory calculations, and CO chemisorption measurements, we establish a direct correlation between the monolayer Pt nanocluster size and the number of interfacial perimeters on Pt/α-MoC$_{1-x}$ catalysts. In this work, these findings provide key insights into the active site configuration of Pt/α-MoC$_{1-x}$ catalysts and open pathways for innovative catalyst design, with the interfacial perimeter identified as a crucial factor in enhancing catalytic performance.

Noble metal-supported catalysts on transition-metal carbides (e.g., MoC, WC, TiC) have been shown to exhibit high activity across a wide range of chemical reactions, including the low-temperature water-gas shift (LTWGS) reaction[1–11], methanol steam reforming[12–16], and CO/CO$_2$ hydrogenation[17–20]. This high activity is largely attributed to the strong metal-support interactions (SMSI), which enhance the dispersion of metal species. In addition, the strong interaction at the metal-carbide interface leads to unique geometric and electronic structures that significantly enhance the activity of the interfacial sites[2,5,13]. Transition-metal carbides share a similar electronic density of states and crystal structure with noble metals, enabling strong electronic interactions and firm binding with precious metals. As a result, noble metals can present as single atoms, monolayer islands, polyhedral nanoparticles, or clusters[1,4,6,14]. Beyond merely anchoring noble metals, carbide supports also actively participate in reactions by activating reactant molecules[5,15].

Zhang et al. demonstrated that Pt/α-MoC$_{1-x}$ catalysts with high metal loading exhibit superior activity and stability compared to those with low metal loading. After 10 h of reaction, the activity of the 0.02% Pt/α-MoC$_{1-x}$ catalyst nearly diminished, while the 2% Pt/α-MoC$_{1-x}$ catalyst maintained its activity[1]. This suggests that the initial activity comes from two types of active sites: those on the α-MoC$_{1-x}$ support and those at the Pt−molybdenum carbide interface. Over time, the α-MoC$_{1-x}$ support sites deactivate due to oxidation, while the platinum−carbide interface remains stable and highly active. This hypothesis is supported by both experimental studies on metal/α-

[1]State Key Laboratory of Heavy Oil Processing, China University of Petroleum, Qingdao, Shandong, P.R. China. [2]Department of Chemical Engineering, Norwegian University of Science and Technology, Trondheim, Norway. ✉e-mail: liuyibin@upc.edu.cn; xiangfeng@upc.edu.cn; de.chen@ntnu.no; yangch@upc.edu.cn

$MoC_{1-x}$ catalysts and theoretical calculations on metal/α-MoC-111 and α-MoC-100 models. Ma's group further confirmed that low-loading metal/α-$MoC_{1-x}$ catalysts experience significant oxidation of the α-$MoC_{1-x}$ surface, unlike high-loading catalysts[1,6,13]. Theoretical and experimental research has consistently shown that the metal-α-$MoC_{1-x}$ interfacial active sites are more reactive than the α-$MoC_{1-x}$ surface alone[2,4,21]. Jin et al. improved the dispersion of Au particles by adjusting the carbonization temperature, thereby increasing the number of active sites at the Au−α-$MoC_{1-x}$ interface and enhancing catalytic activity[4]. Upon activation of passivated Au/α-$MoC_{1-x}$ catalysts, structural reorganization of Au species on the α-$MoC_{1-x}$ surface expanded the Au/α-$MoC_{1-x}$ interfacial perimeter, further boosting catalytic performance[21]. These findings suggest that the interface perimeter is closely linked to catalytic activity, and previous studies have shown that adjusting the size of metal particles is a key strategy for expanding the interfacial perimeter[22,23].

Typically, techniques such as adsorption pulse experiments using probe molecules or transmission electron microscopy (TEM) characterization are employed to determine the particle size of metal species in supported catalysts[24–26]. However, in Metal/α-$MoC_{1-x}$ catalyst systems, the metal species are often distributed as single atoms or atomic layers, making it difficult to accurately determine their particle size using TEM images. CO-pulse experiments are crucial for testing the dispersion of precious metals, but both surface Mo sites and metal sites can adsorb CO molecules, complicating the determination of metal dispersion. Researchers have also correlated catalytic activity with the coordination number of metals using EXAFS characterization, although this method does not directly determine metal particle size[1,6]. The quantitative analysis of active sites is still an important challenge in the Metal/α-$MoC_{1-x}$ catalyst system.

In this study, we quantitatively analyzed the size of Pt clusters and the number of active sites on Pt/α-$MoC_{1-x}$ catalysts with varying metal loadings using a combined experimental and DFT approach. Aberration-corrected high-angle annular dark-field scanning transmission electron microscopy was employed to directly observe the microstructure of the Pt atomic layer and the platinum−molybdenum carbide interface. XPS and XAFS analyses further elucidated the chemical bonding environment of Pt species at the interface. Density Functional Theory (DFT) calculations provided deeper insights into the electronic structure and bonding interactions between Pt atoms and the α-$MoC_{1-x}$ surface. As a result, we were able to clearly identify and quantitatively describe the atomic structure of active sites at the platinum−molybdenum carbide interface for low-temperature WGS reactions. By fine-tuning the size of the atomically thin Pt layer, we optimized the number of active sites, leading to enhanced catalytic performance.

## Results

As illustrated in Fig. S1, the research concept of this study was outlined. In the first step, a series of $Pt_n$/α-MoC-111 and $Pt_n$/α-MoC-100 models (representing the microstructures of Pt atoms on the α-MoC-111 and α-MoC-100 surfaces, respectively) with varying cluster sizes were constructed in DFT based on Pt loading rules. The "sacrificial amount of CO adsorption per Pt atom" parameters for different models were determined according to the CO adsorption rules. In addition, the "sacrificial amount of CO adsorption per Pt atom" parameters of the catalysts were experimentally determined using CO-pulse experiments, establishing a meaningful link between the models and real catalysts. The microstructure models of catalysts with different loadings were further refined using EXAFS characterization. In the second step, the total number of active sites in the catalyst was calculated by identifying the active sites and counting Pt clusters. In the third step, the mass activities of catalysts with different loadings were correlated to the total number of active sites, establishing a benchmark for active

site determination and providing a guide for optimizing catalyst structure to maximize the number of active sites.

## Geometric structure of Pt/α-$MoC_{1-x}$ catalysis

A series of Pt/α-$MoC_{1-x}$ catalysts with Pt loadings ranging from 0.2 to 2.0% were prepared by carbonizing Pt/$MoO_3$ precursors. The influence of Pt loading on the post-carbonization structure of α-$MoC_{1-x}$ was investigated. As shown in Fig. 1a, XRD characterization revealed four characteristic peaks of α-$MoC_{1-x}$ crystals (PDF #08-0384) at 36.9°, 42.6°, 62.3°, and 74.5°[12,16,27], alongside a peak at 39.5° corresponding to β-MoC[12,13]. Further analysis using high-resolution transmission electron microscopy (HR-TEM) and aberration-corrected high-angle annular dark-field scanning transmission electron microscopy (AC-HAADF-STEM) (Figs. S5 and 1d1, e4) showed that the exposed crystal planes of the α-$MoC_{1-x}$ support are the (111) and (200) planes[4,6,15,27,28]. The Pt species were uniformly dispersed, with no visible particles (Fig. S6). On the 0.2% Pt/α-$MoC_{1-x}$ catalyst, Pt primarily exists as single atoms and small clusters (Fig. 1d1–d3). The line-profile intensity of the isolated atoms (Figs. 1d4, e4, S7 and S8) corresponds to Pt atom sizes, indicating that the small clusters form an atomic monolayer on the α-$MoC_{1-x}$ surface, rather than being embedded within the α-$MoC_{1-x}$ structure (details in Supplementary Note 1). In the 1.0% Pt/α-$MoC_{1-x}$ catalyst, Pt also shows a monolayer distribution, resembling a "bamboo raft" arrangement, but with significantly larger clusters than in the 0.2% Pt/α-$MoC_{1-x}$ sample. Isotherm adsorption and desorption curves of the four Pt/α-$MoC_{1-x}$ catalysts exhibit H3-type hysteresis loops (Fig. 1b), with BET surface areas ranging from 71.11 to 76.71 $cm^3 \cdot g^{-1}$. In summary, while adjusting Pt loading does not alter the crystal structure, BET surface area, or pore size distribution of the α-$MoC_{1-x}$ support, it does affect the morphology of the Pt metal.

## Chemical environment of the platinum−molybdenum carbide interface

The chemical environment of Pt and the α-$MoC_{1-x}$ support was characterized using XPS, XAFS, and DFT calculations. As shown in Figs. 2a–c and S9a, XPS analysis of the Pt/α-$MoC_{1-x}$ catalysts surface revealed Pt 4$f$ peaks indicative of Pt species close to metallic $Pt^0$. However, the binding energy of Pt species shifted slightly toward higher energy compared to metallic $Pt^0$ (71.4 eV vs. 71.2 eV), suggesting partial electron transfer from Pt atoms to the α-$MoC_{1-x}$ support, consistent with a strong metal-support interaction. This strong interaction has been confirmed in previous experimental and theoretical studies[1,2,4,7,13,21]. Mo 3$d$ XPS spectra (Fig. 2b) exhibited peaks at 228.3 eV and 231.45 eV, corresponding to $Mo^{\delta+}$ (δ = 0-4) species[12–14,27–30]. The C 1$s$ spectra (Fig. S9a) displayed peaks at 284.8 eV and 283.1 eV, corresponding to C-C/C=C and Mo−C bonds, respectively[6,14,15,21]. Importantly, varying Pt loading did not significantly alter the chemical environment of the Pt/α-$MoC_{1-x}$ catalysts, as confirmed by previous studies that found metal loading changes primarily affect the metal's binding energy in cases of large loading differences[12,14,15,17,28].

XAFS characterization further supported this, showing that the valence states of Pt remained consistent across different Pt loadings (Fig. 2d, e)[1,13,14,17]. The XANES spectra of the three catalysts, normalized at the Pt $L_3$-edge, were similar, indicating that Pt species across all samples are in a comparable electron-deficient but near-neutral state, consistent with the XPS findings. In addition, DFT calculations provided insight into the electronic properties of various $Pt_n$/α-$MoC_{1-x}$ models. The d-band centers are often used to determine reaction activity[31–35]. The density of states (DOS) and partial densities of states (PDOS) of Mo atoms in $Pt_1$/α-MoC-111, $Pt_4$/α-MoC-111, $Pt_8$/α-MoC-111, and $Pt_{12}$/α-MoC-111 models (Figs. 2f and S9b) showed that the d-band center of Mo does not shift significantly with different Pt cluster sizes, suggesting that the intrinsic activity of Mo active sites is unaffected by

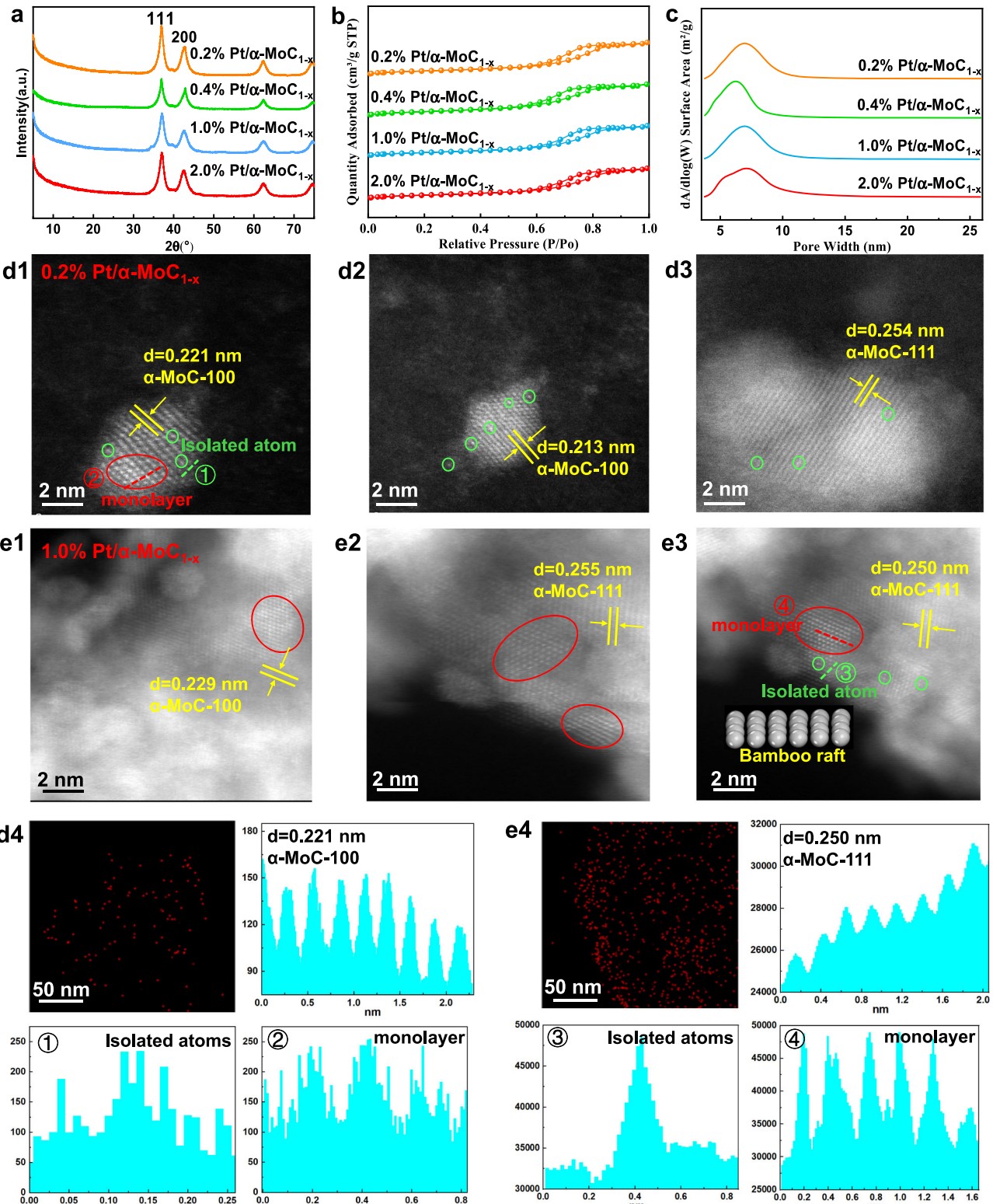

**Fig. 1 | Textural properties of the Pt/α-MoC$_{1-x}$ catalysts with varying Pt loadings: 0.2%, 0.4%, 1.0%, and 2.0%. a** XRD diffraction patterns. **b**, **c** N$_2$ adsorption-desorption isotherms, and pore size distribution profiles for the four catalysts. **d1**–**d3** AC-HAADF-STEM images of 0.2% Pt/α-MoC$_{1-x}$ catalyst. **d4** Line-profile intensities of Pt atomic layers corresponding to panel d1. **e1**–**e3** AC-HAADF-STEM images of 1.0% Pt/α-MoC$_{1-x}$ catalyst. **e4** Line-profile intensities of Pt atomic layers corresponding to panel e3. Pt$_1$ species are highlighted with green dashed circles, and Pt clusters (Pt$_n$) are highlighted with red ellipses.

Pt cluster size[2,8,13,14,36,37]. Similar results were observed in the Pt$_n$/α-MoC-100 models (Fig. S9c, d).

Moreover, Bader charge analysis of Mo atoms at the interface sites across different models (Pt$_1$/α-MoC$_{1-x}$, Pt$_4$/α-MoC$_{1-x}$, Pt$_8$/α-MoC$_{1-x}$, and three variants of Pt$_{24}$/α-MoC$_{1-x}$ with single, double, and triple layers) revealed small charge variations (−0.70 eV to −0.76 eV), indicating minimal differences in the intrinsic activity of the interface sites (Fig. S10). Both experimental results and DFT calculations

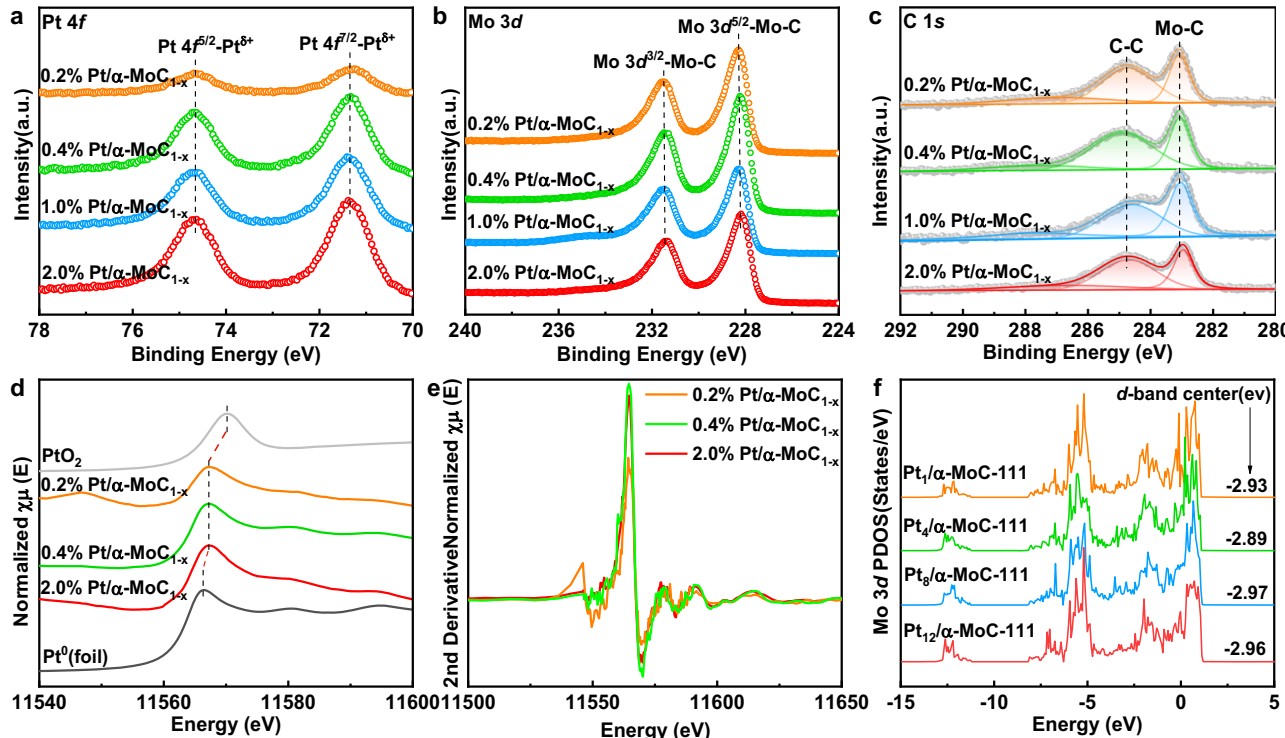

**Fig. 2 | The chemical properties of four Pt/α-MoC₁₋ₓ catalysts. a** In situ XPS spectra of Pt 4*f*. **b** In situ XPS spectra of Mo 3*d*. **c** In situ XPS spectra of C 1*s*. **d** Pt L₃-edge XANES spectra for the Pt/α-MoC₁₋ₓ catalysts. **e** 2nd derivative normalized Pt L₃ XANES. **f** Partial densities of states (PDOSs) for the Mo atoms of Ptₙ/α-MoC-111 models. The Fermi level is set to zero.

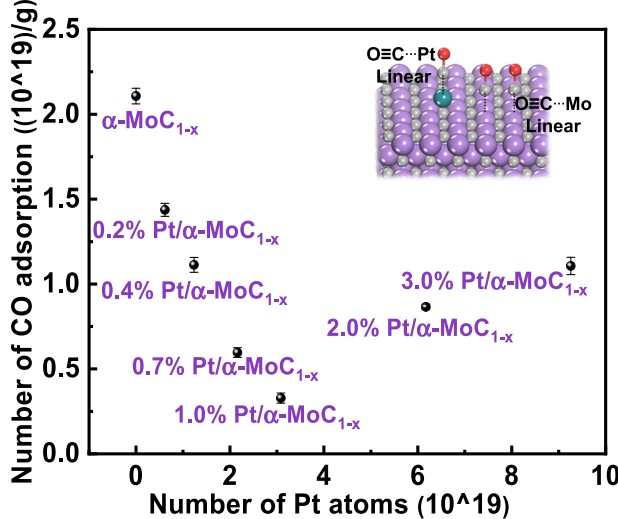

**Fig. 3 | CO adsorption capacity of Pt/α-MoC₁₋ₓ catalysts.** The coordinates of each data point correspond to the values listed in Table S3. Experimental data were obtained from CO-pulse measurement.

demonstrate that Mo active sites and Pt atoms exhibit similar chemical environments across Pt/α-MoC₁₋ₓ catalysts with relatively low Pt loadings.

## Quantitative analysis of Pt−α-MoC₁₋ₓ interface perimeter

The pulse titration method is commonly used to characterize the dispersion of metals on supported catalysts. In this method, probe molecules (such as H₂, O₂ and CO, etc.) are adsorbed onto the metal surface via chemical bonding[38–40]. The metal dispersion and active surface area are then calculated based on the chemical adsorption capacity of these probe molecules. A critical requirement of the pulse titration method is that the probe molecules should not adsorb onto the support material. However, in the Pt/α-MoC₁₋ₓ catalytic system, multiple studies have confirmed that the α-MoC₁₋ₓ support itself also adsorbs CO[2,6,13,17]. Cai et al. addressed this by subtracting the CO adsorption of the α-MoC₁₋ₓ support from the total CO adsorption of the Pt/α-MoC₁₋ₓ catalyst to isolate the CO adsorption attributable to the Pt active sites[12]. However, the introduction of Pt not only increases CO adsorption on Pt active sites but also possibly reduces CO adsorption on Mo active sites, as Pt atoms cover a portion of the Mo sites. Accurately determining metal dispersion remains a challenge, and the measurement of interface site numbers is still lacking.

The CO adsorption on Pt/α-MoC₁₋ₓ catalysts with different Pt loadings was measured by CO-pulse experiments (see details in Supplementary Note 2). As shown in Fig. 3 and Table S3, the CO adsorption first decreases and then increases with increasing Pt loading, reaching its lowest point on the 1.0% Pt/α-MoC₁₋ₓ catalyst. By comparing the CO adsorption on Pt/α-MoC₁₋ₓ catalysts with different loadings to that on α-MoC₁₋ₓ alone, we calculated the change in the number of CO molecules adsorbed per added Pt atom. The sacrificial amount of CO adsorption per Pt atom for the 0.2%, 0.4%, 0.7%, 1.0%, 2.0%, and 3.0% Pt/α-MoC₁₋ₓ catalysts was found to be −0.99, −0.78, −0.70, −0.56, −0.20, and −0.16, respectively. To understand the relationship between CO adsorption variation and the microstructure of Pt/α-MoC₁₋ₓ, we explored the microstructure of Pt clusters on α-MoC₁₋ₓ and the adsorption configuration of CO using DFT calculations.

According to the characterization results and previous reports[1,4,5], the main exposed crystal planes of α-MoC₁₋ₓ are α-MoC-111 and α-MoC-100. These two crystal planes are commonly used as models for DFT calculations of metal/α-MoC₁₋ₓ catalysis[2,13,14,37]. The α-MoC-111 model and α-MoC-100 surface were then also modeled in this work (Fig. S4).

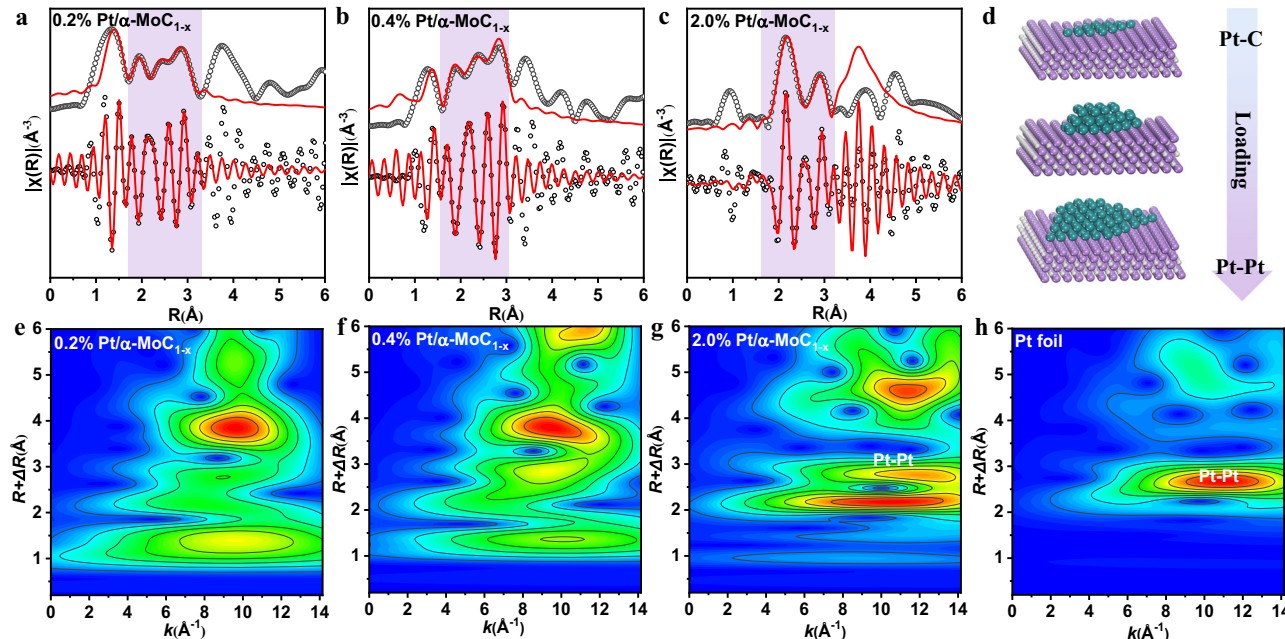

**Fig. 4 | Pt L₃-edge EXAFS (points) and curve fit (line). a** 0.2% Pt/α-MoC₁₋ₓ, **b** 0.4% Pt/α-MoC₁₋ₓ, and **c** 2.0% Pt/α-MoC₁₋ₓ catalysts in R-space, where R represents the distance between the scattering neighbor atoms and the absorbing atom, without correcting for scattering phase shift; the green shaded areas indicate the fitting region. **d** Schematic diagram illustrating the microstructural evolution of the catalyst as the Pt loading increases. **e–h** Wavelet transform (WT) for 0.2% Pt/α-MoC₁₋ₓ (**e**), 0.4% Pt/α-MoC₁₋ₓ (**f**), 2.0% Pt/α-MoC₁₋ₓ (**g**), and Pt foil (**h**).

TEM study (Fig. 1) revealed that single atom, monolayer, clusters, and mano particles of Pt atom exist on the surface, depending on the Pt loading. It is confirmed by the Wavelet transform (WT) analysis of EXAFS spectra (Fig. 4). For the low Pt loading such as 0.2% Pt/α-MoC₁₋ₓ and 0.4% Pt/α-MoC₁₋ₓ, only Pt–C and Pt–Mo bond was detected but no Pt–Pt bond was found, revealing atomically dispersed Pt. For 2.0% Pt/α-MoC₁₋ₓ, Pt–C, Pt–Mo, and Pt–Pt bonds were observed, indicating atomically dispersed and Pt clusters coexisted[1].

The fine structures of the Pt atoms in the Pt/α-MoC₁₋ₓ catalysts were obtained by fitting the extended XAFS oscillation as shown in Fig. 4a–c (the k-space and q-space fitting results are shown in Fig. S19) and results such as the coordination numbers and bond lengths are summarized in Table S6. The data fitting provided equally good fitness for both Pt/α-MoC-111 and Pt/α-MoC-100 structures, where the structures of the samples obtained from DFT were used in the EXAFS curve fitting. A good fitting indicates that the structure established represents well the sample structure, but it is impossible to distinguish α-MoC-111 and α-MoC-100, possibly due to the coexistence of both structures as revealed by TEM. For the 0.2% Pt/α-MoC₁₋ₓ and 0.4% Pt/α-MoC₁₋ₓ catalysts, the atoms surrounding of Pt atom are C atoms or Mo atoms, and coordination depends on the α-MoC structure. The Pt–C bond length is about 1.969 Å, while the Pt–Mo bond lengths are about 2.576 Å and 2.854 Å for Pt/α-MoC-111 and Pt/α-MoC-100 (Table S6), respectively. No Pt–Pt coordination was observed in the first shell layer fit. This implies that the predominant state of Pt is a single atom or monolayer. However, not only Pt–Mo coordination and Pt–C coordination but also Pt–Pt coordination appeared on the 2.0% Pt/α-MoC₁₋ₓ catalyst, implying that the morphology of Pt was newly added to clusters or polyatomic layers. Further, it can be seen from the wavelet transform diagram that the Pt–Pt bond is gradually revealed as the amount of load increases. This conclusion is consistent with the characterization results of Zhang et al., where Pt single atoms were mainly present in the 0.2% Pt/α-MoC₁₋ₓ catalyst, while obvious multilayered Ptₙ nanoclusters appeared in the 2.0% Pt/α-MoC₁₋ₓ catalyst.

As shown in Fig. S11a, b, Pt₄ atom nanocluster and Pt₈ nanocluster were placed on the α-MoC-111 model. After optimization, Pt atoms lay flat on the α-MoC-111 surface in a monolayer pattern. The Pt atoms are arranged regularly at the fcc-C positions. It rationalizes the experimental observation that the Pt nanoparticles automatically resembled atomically dispersed Pt during the carbonation of MoO₃. The two configurations of Pt₁₂ atom nanoclusters were also compared, and it revealed that the larger Pt atoms (10 Pt atoms) bonding with the α-MoC-111 surface is more stable with more negative single-point energy than the structure with less Pt atoms (9 Pt atoms) bonded to the surface (Fig. S11c). The formation of Pt–Pt bonds within the Pt monatomic layer was also examined by comparing the single-point energies of two Pt₂ atom nanocluster models (Fig. S11d), which showed similar energies. This indicates that the interaction between Pt atoms and the α-MoC-111 surface is significantly stronger than the interaction between Pt atoms within the monatomic layer. Similarly, the adsorption pattern of Pt atoms on the α-MoC-100 crystal surface was determined, as illustrated in Fig. S12a, b, where Pt atoms preferentially adsorb at the top-C position. A comparison of the two configurations in Fig. S12a and Fig. S12c shows that multiple Pt atoms bonded to the surface at the top-C position correspond to the more stable conformation. In Fig. S12c, continuous Pt atom nanoclusters are more stable than discontinuous ones. Based on the experimental results, the loading rules are defined which refer to guidelines that describe how metal Pt interacts with the surface of molybdenum carbide, including the binding sites and configurations. These rules ultimately determine the presentation state of Pt on the surface, whether as single atoms, single atomic layers, particles, or clusters. In summary, the loading rules for Pt atoms on the α-MoC-111 and α-MoC-100 surfaces are as follows: (1) Pt atoms preferentially adsorb at the fcc-C position on the α-MoC-111 surface and at the top-C position on the α-MoC-100 surface; (2) The Ptₙ configurations on both surfaces tend to form a single, continuous layer rather than a bilayer or multilayer structure; (3) Pt atoms in the monolayer are more likely to interact with the

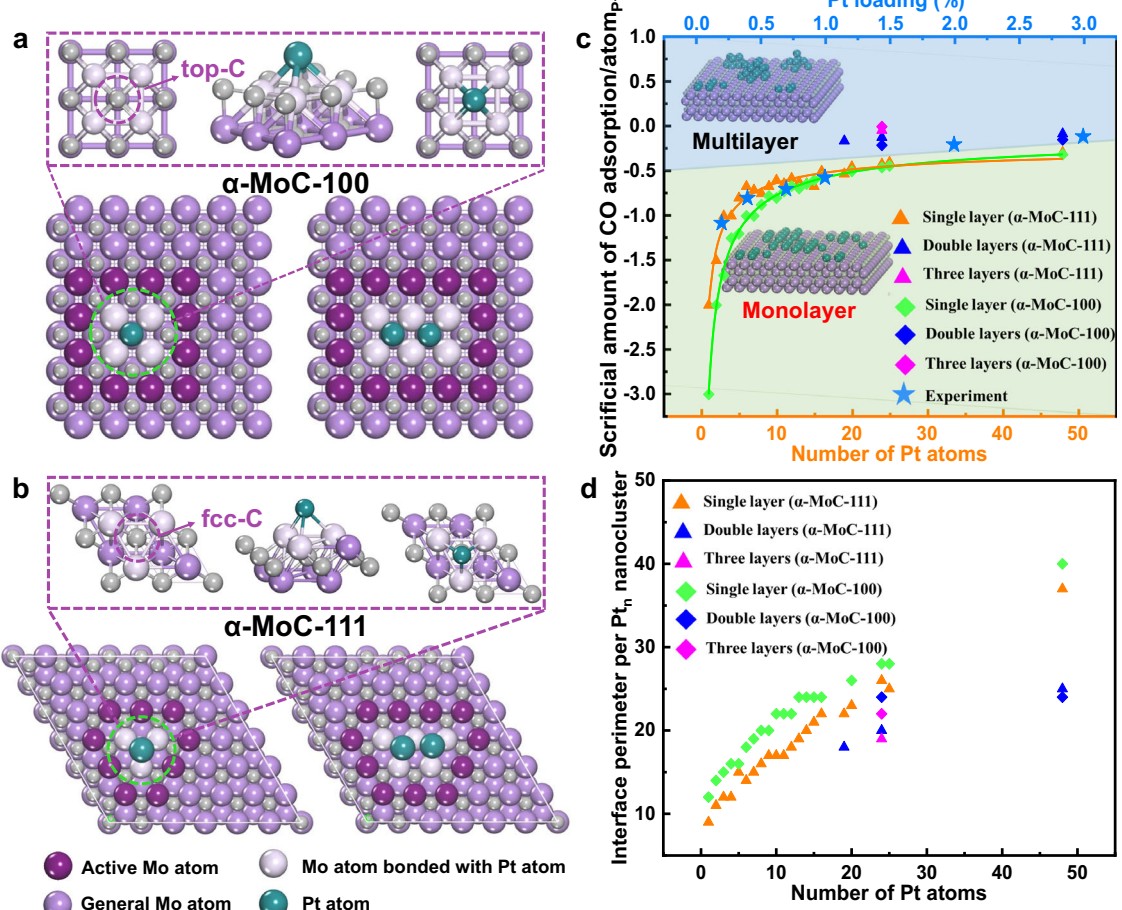

**Fig. 5 | Simulations of the Pt$_n$ nanocluster on the α-MoC-100 surface and α-MoC-111 surface. a** The structure of Pt$_1$ and Pt$_2$ on the α-MoC-111 surface. **b** The structure of Pt$_1$ and Pt$_2$ on the α-MoC-100 surface. Mo atoms were classified into three categories due to the influence of Pt atom: Mo atoms bonded with Pt atom, general Mo atoms, and Mo atoms of Pt-α-MoC$_{1-x}$ interface (also defined as active site Mo atoms). **c** The relationships between the number of Pt atoms of the Pt$_n$ nanocluster models (or the Pt loading of m% Pt/α-MoC$_{1-x}$ catalyst) and sacrificial amount of CO adsorption per Pt atom. **d** The functional relationship between interface perimeter and Pt cluster size.

α-MoC-111 and α-MoC-100 surfaces in a continuous, raft-like arrangement.

The adsorption behavior of CO molecules at Pt and Mo sites was systematically studied to establish the "CO adsorption rules" on Pt/α-MoC$_{1-x}$ surfaces. These rules define the preferred adsorption sites and configurations of CO molecules, which vary based on the catalyst type. As shown in Fig. S14a, on the α-MoC-111 surface, CO molecules adopt the O≡C...Mo configuration, with an adsorption energy of −1.68 eV. When two CO molecules are adsorbed on the α-MoC-111 model, each CO molecule binds independently to adjacent Mo sites, resulting in identical adsorption configurations without repulsion between the CO molecules. In the Pt$_4$/α-MoC-111 model, the mono-coordinated CO adsorption configuration at the Pt site is more stable than the tri-coordinated one. On the α-MoC-100 surface, two CO molecules can adsorb linearly on adjacent Mo sites, as depicted in Fig. S14b. Although the initial configuration considered multi-coordinated CO adsorption on Pt, optimization revealed that CO prefers to adsorb in a linear fashion. As shown in Fig. S15, the adsorption configurations of CO molecules at both Pt-α-MoC$_{1-x}$ interface and non-Pt-α-MoC$_{1-x}$ interface sites were optimized. The adsorption energies on the non-Pt-α-MoC$_{1-x}$ interface configuration is more negative than that on the Pt-α-MoC$_{1-x}$ interface configuration. The results indicate that CO molecules migrate from Pt-α-MoC$_{1-x}$ interface sites and preferentially adsorb on adjacent sites of the interface. Meanwhile, at non-Pt-α-MoC$_{1-x}$ interface sites, CO molecules stably adsorb onto Mo sites. This conclusion is also

supported by the adsorption energy comparison at different sites, as shown in Fig. S13.

Based on the DFT and experimental results, CO adsorption rule was established, which refers to the guidelines governing the adsorption sites and configurations of CO molecules on Pt/MoC$_{1-x}$ surfaces. The CO adsorption rules on Pt$_n$/α-MoC-111 and Pt$_n$/α-MoC-100 structures can be summarized as follows: (1) CO molecules adsorb linearly on adjacent Mo sites, with each Mo site binding to one CO molecule; (2) CO molecules also adsorb linearly on Pt sites, with each exposed Pt atom binding to one CO molecule; and (3) Mo atoms coordinated with Pt lose the ability to adsorb CO molecules.

Based on the Pt loading rules, Pt$_n$/MoC-111 and Pt$_n$/MoC-100 structures ($n$ = 2−25) were established, as shown in Figs. S16 and S17. For the Pt$_{24}$ atom nanocluster, monolayer, bilayer, and three-layer models were constructed. The interface structures of Pt$_1$ and Pt$_2$ on α-MoC-100 and α-MoC-111 surfaces are illustrated in Fig. 5a, b. These figures depict the adsorption sites of Pt atoms, with Pt atoms preferentially adsorbing at the top-C site on α-MoC-100 and the fcc-C site on α-MoC-111. Figure 5c shows Mo sites bonded to Pt at the interface, which reduces CO adsorption capacity. The number of these sites increases with higher Pt loading.

Using the interface structures shown in Figs. S16 and S17, the sacrificial amount of CO adsorption for each cluster was determined by counting the Mo sites bonded with Pt at the interface. The sacrificial amount of CO adsorption per Pt atom is defined as the loss of CO

adsorption sites per Pt atom in the cluster, calculated by normalizing the total loss of CO adsorption sites by the total number of Pt atoms in the cluster. A negative value indicates a loss of CO adsorption sites. According to the CO adsorption rules, changes in the number of adsorption sites at Mo and Pt sites were calculated for each model, as shown in Table S4 and Table S5. For monolayer-distributed Pt atoms on the α-MoC-111 surface, the sacrificial amount of CO adsorption per Pt atom decreases from -2 to -0.29 as the number of Pt atoms increases from 1 to 48 (Table S4). Similarly, for the α-MoC-100 surface, the sacrificial amount decreases from −3 to −0.31 with the same increase in Pt atoms (Table S5). The sacrificial amount of CO adsorption per Pt atom was plotted against the Pt cluster size, as shown in Figs. S16 and S17. In multilayer nanocluster models, the sacrificial amount of CO adsorption per Pt atom decreases to around 0.1 in bilayer models and nearly 0 in the three-layer model (Pt$_{24}$/α-MoC-100-three layers).

Figure 5c presents the functional relationships between Pt cluster size and the sacrificial amount of CO adsorption per Pt atom for monolayer Pt$_n$/α-MoC-111 and Pt$_n$/α-MoC-100 models. The relationships were fitted to the equations:

$$\text{For } \alpha - \text{MoC} - 111 : y = -16.23/(1 + 9.09x^{0.69}) - 0.065$$

$$\text{For } \alpha - \text{MoC} - 100 : y = -97.97/(1 + 540.54x^{0.63}) - 0.20.$$

For the same number of Pt atoms, the sacrificial amount of CO adsorption per Pt atom in multilayer models is greater than in monolayer models, indicating that the data points for multilayer models deviate from the fitted curve for monolayer models as shown in Fig. 5c. This parameter bridges the Pt loading content and the size of Pt$_n$ nanoclusters. The average size of Pt$_n$ nanoclusters can be determined from CO-pulse experiments by calculating the sacrificial amount of CO adsorption per Pt atom. For example, the 0.2% Pt/α-MoC$_{1-x}$ catalyst corresponds to Pt$_4$ monolayer nanoclusters on the α-MoC-111 surface and Pt$_6$ monolayer nanoclusters on the α-MoC-100 surface. Table S7 shows the average cluster sizes for 0.4%, 1.0%, 2.0%, and 3.0% Pt/α-MoC$_{1-x}$ catalysts. Pt loadings below 1% exhibit monolayer dispersion, while loadings above 1% result in multilayer nanoclusters. This trend aligns with the findings of Zhang et al. and Lin et al., who reported that multilayer Pt$_n$ nanoclusters form as the loading increases[1,14]. Figure 5d shows that monolayer configurations have larger interface perimeters compared to multilayer structures, as illustrated by plotting the interface perimeter per Pt nanocluster against the number of Pt atoms. This figure highlights how Pt configurations on α-MoC$_{1-x}$ surfaces influence CO adsorption efficiency and catalytic interface structure. These findings demonstrate that using the sacrificial amount of CO adsorption per Pt atom to estimate Pt cluster size is a reasonable approach.

## Pt-α-MoC$_{1-x}$ interface perimeter

Previous studies have shown that surface Mo atoms influenced by Pt atoms act as active sites for catalyzing water-gas shift reactions[5,41]. For example, Li et al. demonstrated that in Ir/α-MoC$_{1-x}$ catalysts, Ir atoms act as promoters, enhancing the activity of the system despite not directly participating in the reaction[5]. Our previous work also showed through DFT calculations that Pt-influenced Mo sites exhibit improved activity and stability[41]. In Fig. 5a, b, the active sites at the Pt-α-MoC$_{1-x}$ interface on α-MoC-111 and α-MoC-100 surfaces are highlighted (marked in dark violet). The interface perimeter, representing the catalyst's active sites, is quantified by defining one atom as a unit of perimeter. We calculated the average sizes of Pt$_n$ nanoclusters in the m% Pt/α-MoC$_{1-x}$ catalysts and their respective interface perimeters from the microscopic models shown in Figs. S16 and S17.

As shown in Fig. 6d, the interface perimeter increases with cluster size. We calculated the total perimeter for each catalyst by multiplying the number of Pt$_n$ nanoclusters by their corresponding interface perimeter. Figure 5d and Table S7 present the Pt-α-MoC$_{1-x}$ interface perimeters for the α-MoC-111 and α-MoC-100 models. To validate the accuracy and rationality of our model construction, we correlated the interface perimeter with actual reaction activity. A monoclinic increase in mass activity with the interface perimeter site number per gram of catalyst (Fig. 6d), giving a benchmark of our method for measuring the active sites.

## Low-temperature WGS reaction activity

The activated catalysts were evaluated for the low-temperature water-gas shift (LTWGS) reaction at 100–200 °C and 1.0 atm. As shown in Fig. 6a, all m% Pt/α-MoC$_{1-x}$ catalysts exhibited high activity for the low-temperature WGS reaction. Above 160 °C, the 1.0% Pt/α-MoC$_{1-x}$ catalyst easily achieved thermodynamic equilibrium conversion at a high space velocity of 90,000 mL·g$^{-1}$·h$^{-1}$. The mole-specific activity (mol$_{H_2}$·mol$_{Pt}^{-1}$·s$^{-1}$) and mass activity (μmol·g$_{cat}^{-1}$·s$^{-1}$) of Pt/α-MoC$_{1-x}$ catalysts with different Pt loadings, measured under differential conditions at 150 °C, are summarized in Fig. 6b. As Pt loading increased, the mass activity showed a continuous upward trend. The specific reaction rate, however, first increased and then decreased, with the 0.4% Pt/α-MoC$_{1-x}$ catalyst achieving the highest Pt atom utilization efficiency.

The Arrhenius plot of reaction rate versus temperature is presented in Fig. 6c, with apparent activation energies (Table S8) showing a similar trend of first decreasing and then increasing. The 1.0% Pt/α-MoC$_{1-x}$ catalyst displayed the lowest apparent activation energy at 51.0 kJ·mol$^{-1}$. Figure 6d graphically represents the mass activities of the Pt/α-MoC$_{1-x}$ catalysts with different Pt loadings plotted against the interfacial perimeter number, as derived from the model. Two key insights emerge from this: first, the Pt-α-MoC$_{1-x}$ interface perimeter is confirmed as the critical active site for the low-temperature WGS reaction, and second, the validity of the model is reinforced. As discussed previously, XPS, XANES, and DFT calculations indicate that platinum loading does not significantly affect the chemical environment of the Pt and Mo. It is well-established that the chemical environment of the catalyst surface is crucial in influencing catalytic reactivity[42–47]. Therefore, the reactivity of the Pt-α-MoC$_{1-x}$ interface across different cluster sizes is considered to be equivalent.

It is important to note that the apparent activation energies of the catalysts vary. This variation arises because the apparent activation energy reflects not only the activation energy barrier of the intrinsic active sites but also the combined effect of the activation energy barrier and the adsorption energy on all active sites.

As shown in Fig. 6d, whether the (111) or (100) crystal surface is used as the active component, the mass activities and interfacial perimeters of the models exhibit an excellent linear relationship. This suggests that the method for quantitatively characterizing the reactive sites is both reasonable and feasible. We believe this method can be extended to other catalyst systems. For catalyst materials where both the support and the metal are active, a probe molecule can be selected to quantitatively analyze the chemisorption properties of both the loaded metal and the support. By separately determining the rules governing metal binding and probe molecule adsorption on the model surface, the variability of chemisorption with metal loading can be characterized. Based on these chemisorption results, the adsorption variability of the probe molecules can be used to retrieve corresponding models from the model library, allowing for the determination of the cluster size and number of active sites of the metals in the catalyst. Finally, the validity of the model can be confirmed by correlating the interfacial perimeter

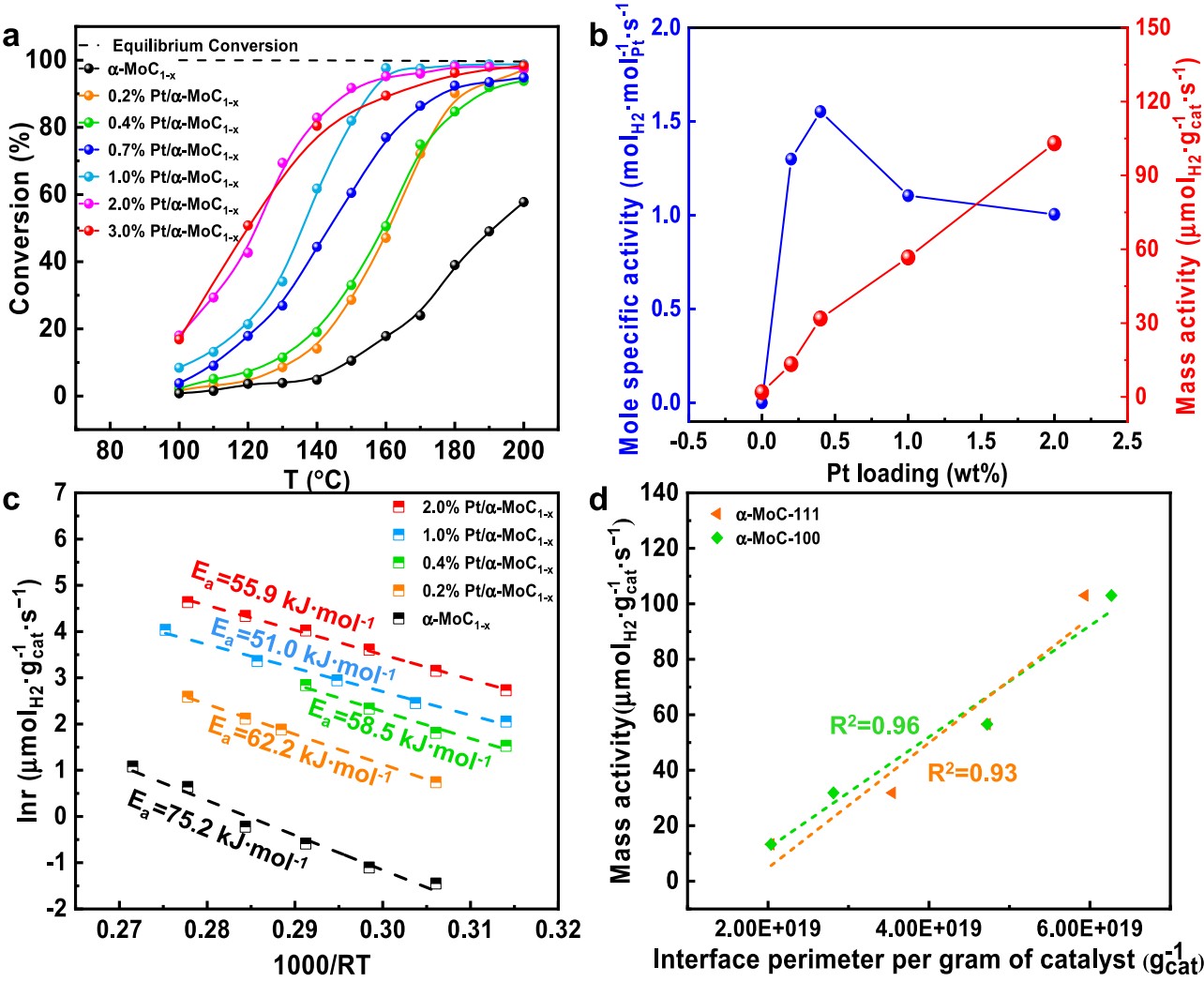

**Fig. 6 | Reactivity of Pt/α-MoC$_{1-x}$ catalysts. a** CO conversion over different catalysts at 100–200 °C: 10.5% CO, 21% H$_2$O, 20%Ar/N$_2$ balance, 90,000 mL·g$^{-1}$·h$^{-1}$. The dashed line indicates the CO thermodynamic equilibrium conversion calculated from Aspen Plus software. **b** The mole-specific activity over m% Pt/α-MoC$_{1-x}$ catalysts (mol H$_2$ per mol Pt per second, left axis, blue) at 150 °C. The mass activity over m% Pt/α-MoC$_{1-x}$ catalysts (mol H$_2$ per unit mass of the catalyst per second, right axis, red) at 150 °C. **c** The Arrhenius plots of m% Pt/α-MoC$_{1-x}$ catalysts. **d** The relationship between the interface perimeter per gram of catalyst and mass-specific activity. The detailed data and calculation methods are referred to Table S7.

derived from the models with the mass reaction activity observed in experiments.

## Discussion

In supported catalysts, the complex and varied surface and interface structures make it difficult to identify and quantify active sites. The quantification of active sites is often closely related to the particle size of the supported metal. On Pt/α-MoC$_{1-x}$ catalysts, many studies have demonstrated that Pt atoms are dispersed on the α-MoC$_{1-x}$ surface in a form of monolayer due to the strong metal-support interaction of Pt-α-MoC$_{1-x}$[1,12,13,17]. It is very difficult to count the cluster size of Pt on the Pt/α-MoC$_{1-x}$ catalysts. A combined DFT calculation and chemisorption method is proposed to quantify Pt cluster size and the number of active sites.

This research has significantly advanced our understanding of Pt/α-MoC$_{1-x}$ catalysts, particularly their remarkable efficacy in low-temperature water-gas shift reactions. Despite the inherent challenges in quantitatively characterizing and fine-tuning the active sites of these catalysts, our findings offer insights and a method for the measurement of active site numbers. We discovered that the mass activity of fully exposed monolayer Pt nanoclusters on molybdenum carbides is markedly higher than that of isolated single Pt atoms and the bulk molybdenum carbide catalysts, by an impressive one and two orders of magnitude, respectively, for low-temperature WGS at 100–200 °C. The cornerstone of this achievement lies in the detailed elucidation and quantification of the active sites situated along the Pt-molybdenum carbide interfacial perimeter. By leveraging a synergistic approach that combines the sacrificial amount of CO adsorption per Pt atom, Density Functional Theory (DFT) calculations, and CO chemisorption measurements, we established a direct link between the number of interfacial perimeters and the size of monolayer Pt nanoclusters. This correlation not only sheds light on the intricate active site configuration of Pt/α-MoC$_{1-x}$ catalysts but also paves the way for the design of advanced catalysts. Our research emphasizes the critical role of the interfacial perimeter of fully exposed monolayer clusters in enhancing catalytic performance, thereby offering a paradigm for catalyst development in a wide array of chemical reactions. However, the proposed method is limited to monolayer dispersed catalyst systems. The quantification of particle size and active sites of multilayer atomic clusters and metal particles still needs to be improved on the basis of this method.

## Methods

### Reagents

All chemicals including ammonium heptamolybdate tetrahydrate $(NH_4)_6Mo_7O_{24} \cdot 4H_2O$ and chloroplatinic acid hexahydrate $(H_2PtCl_6 \cdot 6H_2O)$ were purchased from Sinopharm Chemical Reagent Co., Ltd. and used without further purifications. All gases including $NH_3$ (99.999%), He (99.999%), $N_2$ (99.999%), Ar (99.999%), CO (99.999%), 15% $CH_4/H_2$, 1% $O_2/Ar$ and 5% CO/He were purchased from Qingdao Xinkeyuan Gas Co., Ltd. and used without further purifications. Deionized water with a resistance no less than 18.5 MΩ was used.

### Preparation of $MoO_3$ precursor

20.00 g $(NH_4)_6Mo_7O_{24} \cdot 4H_2O$ was subjected to calcination in a muffle furnace at a heating rate of $2\,°C \cdot min^{-1}$ until reaching a temperature of 500 °C, followed by further calcination at this temperature for 4 h to obtain 16.30 g $MoO_3$ powder.

### Preparation of α-$MoC_{1-x}$ support

The α-$MoC_{1-x}$ support was prepared in a micro-fixed-bed reactor using a two-stage temperature-programmed nitridation–carbonization process. The detailed procedure was as follows: 2.00 g $MoO_3$ precursor (40–60 mesh) was placed in the reactor, followed by the introduction of pure $NH_3$ gas ($50\,mL \cdot min^{-1}$). The temperature was then raised to 700 °C at a rate of $5\,°C \cdot min^{-1}$ and maintained for 2 h, resulting in the formation of γ-MoN intermediate. Subsequently, after cooling to room temperature, the reaction gas was switched to a mixture of 15% $CH_4/H_2$ ($50\,mL \cdot min^{-1}$), heated up to 700 °C at a rate of $5\,°C \cdot min^{-1}$, and kept for another 2 h. After carbonization and subsequent cooling to room temperature, α-$MoC_{1-x}$ material was obtained. To address the instability issue associated with α-$MoC_{1-x}$, the atmosphere was changed to one consisting of 1% $O_2/Ar$ ($30\,mL \cdot min^{-1}$) and maintained for ~5–6 h. The final mass of α-$MoC_{1-x}$ was 1.48 g.

### Preparation of m% Pt/α-$MoC_{1-x}$ catalyst (m = 0.2, 0.4, 1.0, 2.0)

Firstly, $Pt/MoO_3$ was prepared using the initial wet impregnation method, followed by carbonization of $Pt/MoO_3$ through temperature-programmed procedures to obtain Pt/α-$MoC_{1-x}$. The specific procedure involved weighing 1.3333 g of $MoO_3$ precursor in a beaker and subsequently dropwise add the 265 μL of $H_2PtCl_6$ (aq, 0.0386 $mol_{Pt} \cdot L^{-1}$), 265 μL of $H_2PtCl_6$ (aq, 0.0772 $mol_{Pt} \cdot L^{-1}$), 265 μL of $H_2PtCl6$ (aq, 0.1930 $mol_{Pt} \cdot L^{-1}$), 265 μL of $H_2PtCl_6$ (aq, 0.3860 $mol_{Pt} \cdot L^{-1}$) to form a slurry-like flow state. After thorough mixing, the mixture was aged at room temperature for 10 h and then dried overnight in an oven at 60 °C. The obtained solid powder was ground and calcined in a muffle furnace at 400 °C for 4 h to obtain m% $Pt/MoO_3$ precursor (where m represents the mass percentage of Pt in Pt/α-$MoC_{1-x}$). Subsequently, m% $Pt/MoO_3$ (40–60 mesh) was placed in a fixed-bed reactor while introducing a mixed gas consisting of 15% $CH_4/H_2$ mixed gas ($50\,mL \cdot min^{-1}$). The temperature was gradually increased to 700 °C at a rate of $5\,°C \cdot min^{-1}$ and maintained for 2 h to achieve carbonization. Finally, m% Pt/α-$MoC_{1-x}$ catalyst was obtained for subsequent reaction evaluation. For characterization purposes, the catalyst underwent passivation by exposure to an atmosphere containing 1% $O_2/Ar$ ($30\,mL \cdot min^{-1}$) for 5–6 h before removal. The final mass of 0.2% Pt/α-$MoC_{1-x}$, 0.4% Pt/α-$MoC_{1-x}$, 1.0% Pt/α-$MoC_{1-x}$ and 2.0% Pt/α-$MoC_{1-x}$ were 0.9812 g, 0.9822 g, 0.9950 g, and 1.0100 g, respectively. The actual yield is lower than the theoretical yield because of some losses due to the adhesion of the cup walls.

### Evaluation of catalytic performance

The catalytic performance of the WGS reaction was evaluated in a gas continuous flow fixed-bed quartz reactor with an inner diameter of 8 mm. The catalyst (0.2 g, 40–60 mesh) was diluted with inert SiC (0.6 g, 40–60 mesh). Before testing, the catalysts were activated in situ by subjecting them to a mixed gas consisting of 15% $CH_4/H_2$ at a space velocity of $50\,mL \cdot g^{-1} \cdot h^{-1}$ at 700 °C for 2 h. The reactant mixture consisted of CO (10.5%), $H_2O$ (21%), and Ar (5%) balanced with $N_2$, resulting in a total flow rate of $300\,mL \cdot min^{-1}$ and a WHSV of $90,000\,mL \cdot g^{-1} \cdot h^{-1}$. Water was injected into an evaporator at 200 °C for gasification, and mixed with CO and $N_2$ gas flows into the reactor. The reacted gas mixture is passed through a condenser to remove unreacted $H_2O$. The gas composition was analyzed by Agilent gas chromatograph. CO conversion and carbon balance of the WGS reaction were calculated quantitatively according to Eq. (1) and Eq. (2), respectively.

$$X_{CO}(\%) = (F_{CO}^{in} - F_{CO}^{out})/F_{CO}^{in} \times 100\% \qquad (1)$$

$$B_C(\%) = (F_{CO_2}^{out} + F_{CO}^{out} + F_{CH_4}^{out})/F_{CO}^{in} \times 100\% \qquad (2)$$

Where $F_{CO_2}^{out}$, $F_{CO}^{out}$, $F_{CH_4}^{out}$, $F_{CO}^{in}$ represent the flow rates of $CO_2$, CO, and $CH_4$ in the outlet reforming flow and the flow rate of CO in the feed steam, respectively. The carbon balance of each reaction evaluation system is within the range of 95–99%.

### WGS kinetic measurements

All kinetic data were acquired under conditions of low conversion and significant deviation from reaction equilibrium (0–20%). The reaction rate, denoted as r ($mol \cdot g_{cat}^{-1} \cdot s^{-1}$), was calculated by Eq. (3). The flow rate of carrier gas $N_2$ was adjusted to ensure a balanced total gas flow.

$$r = \frac{F_{CO} \cdot x_{CO}}{W_{cat}} \qquad (3)$$

### Catalyst characterization

The crystal structures of the catalysts were measured on an X'pert PRO MPD diffractometer to obtain X-ray diffraction patterns (XRD). The specific surface areas were determined using a Micromeritics ASAP 2020 instrument. High-resolution transmission electron microscopy (HR-TEM) images were obtained by using a JEOL JSM-2010 instrument. The morphology of metallic Pt was imaged on an Aberration-correction high-angle annual dark-field scanning transmission electron microscopy (AC-HAADF-STEM) equipped with FEI Themis Z. The catalyst was dispersed in ethanol by ultrasonic for 5–10 min, and the resulting solution was dropped dropwise onto a porous carbon film supported by a copper TEM grid. The composition and valence states of the elements on the fresh catalyst surface were analyzed on an X-ray photoelectron spectroscopy (XPS) instrument. The sample preparation and detection process was conducted under stringent measures to guarantee anaerobic conditions. Firstly, the m% Pt/α-$MoC_{1-x}$ catalysts were prepared in a fixed-bed reactor and subsequently transferred to a glove box filled with argon gas under an inert atmosphere of argon. Prepared the XPS detection samples in the glove box. The samples were subsequently transferred from the glove box to the sample analysis center of the XPS instrument via an in situ transfer vessel. The actual contents of Pt were measured with inductively coupled plasma atomic emission spectroscopy (ICP-AES). To ensure that the samples were not oxidized and the content was accurately measured, a certain amount of aqua regia was used to digest the samples in the glove box before ICP determination. The CO-pulse adsorption analysis was conducted using the Micromeritics Autochem II 2920 chemisorption instrument. The analysis procedure was as follows: The U-shaped tube was filled with 0.1333 g $Pt/MoO_3$ or MoN catalyst and purged with high-purity helium gas for 30 min at 300 °C to eliminate surface adsorbed water and impurities. Subsequently, the temperature was reduced to 50 °C. The gas flow was then switched to a mixture of 15 vol% $CH_4/H_2$ ($20\,mL \cdot min^{-1}$), and the catalyst was heated at

a rate of 10 °C per minute until reaching 700 °C over 2 h. Afterward, the temperature was lowered back to 50 °C, and the system underwent another purge with high-purity helium gas for an additional duration of 30 min to remove any remaining pre-treatment gases. Finally, CO-pulse experiments were conducted at a temperature of 50 °C, and CO concentration was detected by a TCD detector. The chemical environment of metal Pt on 0.2% Pt/α-MoC$_{1-x}$, 0.4% Pt/α-MoC$_{1-x}$, and 2.0% Pt/α-MoC$_{1-x}$ catalysts was measured by the BL14W beamline of Shanghai Synchrotron Radiation Facility (SSRF) using a Lytle detector. Pt foil and PtO$_2$ were used as standards for this measurement. The activated catalyst was sealed between two Kapton windows in the glove box under argon protection to obtain XAFS spectra of fresh catalyst. The XAFS data underwent processing using the Ifeffit package software (version 1.2.11). The extended XAFS oscillations were fitted using FEFF models generated from α-MoC (space group Fm$\bar{3}$m) and Pt (Fm$\bar{3}$m) crystal structures based on the back-scattering equation (Fig. S18). MATLAB 2023a software was used to process the wavelet transform[48,49]. After data processing, k-space spectra, R-space spectra, and continuous wavelet transform figures can be obtained.

## Computational details

All DFT calculations were conducted in the Vienna Ab Initio Simulation Package (VASP). The electron smearing adopted the PBE functional with the first-order Meth–fessel–Paxton method. The projector augmented wave (PAW) method was used to describe the interaction between ion cores and valence electrons. The Brillouin zone adopted a 3 × 3 × 1 k-point Mon–khorst–Pack grids sampling method, which has been proven efficient for the cell under investigation. The cutoff energy of plane wave was 400 eV. Convergence criteria were set such that all forces on atoms were below 0.01 eV·Å$^{-1}$. Calculation parameters underwent convergence tests and were also validated against literature[5,6,15,28]. In this study, Gibbs free energies were corrected by reaction temperature (423 K) and zero point energy (ZPE) (Eq. (4))[50,51].

$$G = E + ZPE - TS \tag{4}$$

The adsorption energies calculated from Eq. (5) were utilized for assessing the adsorption capacities. Generally, negative adsorption energy indicates a strong interaction between the adsorbed molecule and the catalyst model[31,32,52,53].

$$G_{ads} = G_{A+surface} - G_A - G_{surface} \tag{5}$$

where $G_A$, $G_{surface}$, and $G_{A+surface}$ are the energy of the reactant molecule, energy of the surface, and the energy of surface species on the surface, respectively.

## Computational models

Firstly, the α-MoC crystal cell was optimized (Fig. S4a). The lattice constant of the optimized fcc-α-MoC cell is 4.332 Å (approximate to the experimental value: 4.270 Å), with a Mo–C bond length of 2.166 Å and a Mo-Mo bond length is 3.063 Å. As shown in Fig. S4b, c, we constructed models for α-MoC-100 and α-MoC-111 using a 6 × 6 unit cell consisting of seven layers and four layers, respectively. The atoms in the bottom four layers of the α-MoC-111 model and the bottom two layers of the α-MoC-100 model were kept fixed while relaxing the remaining atoms. These surfaces were modeled with a vacuum spacing of -15 Å between successive metal slabs. In addition, Pt$_n$/α-MoC-111 and Pt$_n$/α-MoC-100 models with different Pt cluster sizes were constructed (Figs. S16 and S17).

## Data availability

All relevant data that support the findings of this study are presented in the manuscript and supplementary information file. Atomic

coordinates for all models are provided with this paper and in Supplementary Data 1. Source data are provided in this paper and in Source Data 1. Source data are provided with this paper.

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

## Acknowledgements

This work was supported by the National Natural Science Foundation of China-outstanding Youth Foundation (22322814), National Natural Science Foundation of China (22272199, 22208374), and the Key R&D Program of Shandong Province (2021ZLGX06).

## Author contributions

Ruiying Li: conceptualization, investigation, data analysis, and writing—original draft. Jinyuan Shang: data analysis. Fei Wang: data curation and visualization. Qing Lu: data curation and visualization. Hao Yan: writing–review and editing. Yongxiao Tuo: writing—review and editing. Yibin Liu: supervision and writing—review and editing. Xiang Feng: writing—review and editing. Xiaobo Chen: writing–review and editing. De Chen: supervision, data analysis, and writing—review and editing. Chaohe Yang: writing—review and editing.

## Funding

 Olavs Hospital - Trondheim University Hospital).

## Competing interests

The authors declare no competing interests.
