## [Peer Review File · Nature Communications]

Quantification and optimization of platinum–molybdenum carbide interfacial sites to enhance low-temperature water-gas shift reaction

Corresponding Author: Professor De Chen

Version 0:

Reviewer comments:

Reviewer #1

(Remarks to the Author)

This manuscript highlights the high performance of Pt/MoC for low-temperature water gas shift reaction, which have been found previously. The quantitatively describing and of the active sites at the interface perimeter between Pt and MoC was concentrated on in order to clarify the underlying site configuration. The authors made significant endeavors to recognize the CO adsorption site and correlated it with active sites. However, there are many shortcomings that need further clarification.

(1) It is well known that with the increasing of the Pt loading amount, the metal size also evolved from single atoms to clusters and then nanoparticles. Firstly, the adsorption mode of CO can also change from linear mode to bridged one. Thus, how to precisely quantify the active sites from CO adsorption amount. Moreover, for high loading amount of Pt, there should be a mixture of Pt single atoms and other clusters. It also affects the analysis of the contribution to CO adsorption.

(2) The manuscript considers that the states of Pt and Mo sites are not changed with the loading amount and the Pt size, which is not consistent with conventional conclusion. Particularly, for single atoms, there is coordination between Pt and Mo or C species, while for clusters or particularly nanoparticles, the Pt-Pt bond should exist. The different coordination environment can affect the state of Pt and Mo.

(3) It is reasonable to study the location of Pt species on the face of MoC, particularly it has been found that there is (111) and (100) facet. However, the evidence of the location state is not ample. Which one is more stable, on MoC(111) or (100). Moreover, it needs more evidence to identify the location sites of MoC.

(4) MoC is recognized to be a noble-metal like support that means it can easily adsorb CO or H₂ like noble-metal species. The method of sacrificial CO adsorption per Pt atom may be effective to distinguish the adsorption on Pt and MoC sites. Are there any CO spillover effects that can influence the quantification? If there is no spillover effect, the use of H₂ chemisorption or H₂-O₂ titration experiment can further support these results.

(5) The DFT calculation can exactly study the effects of Pt layers while the experimental evidences are not enough. The AC-HAADF STEM may provide more information, which need more characterization from different perspectives.

Reviewer #2

(Remarks to the Author)

Very interesting study in a hot topic these days. The following points must be addressed by the authors:

- 1) Bottom of page 7: Could the Pt be embedded in the carbide substrate?
- 2) In figure 3c, how stable is the species with C-C bonds?
- 3) Page 11, can the authors rule out adsorption of CO on the Pt-MoC_x interface?
- 4) Top of page 13, the authors need to give a better justification of the models used in the DFT calculations.
- 5) Pages 22-23, the "Discussion" is kind of short and looks like a "Conclusions" section. The article will benefit from a larger and more polished Discussion

Reviewer #3

(Remarks to the Author)

The authors present the synthesis of a series of Pt/MoC_{1-x} with varying Pt loadings and use this series to evaluate a model for particle/interface size and correlate this to catalytic activity for the water gas shift reaction. I think this is an interesting approach, but struggle to see the same novelty as the authors. The paper attempts to address the challenges associated with characterizing Pt-based sites on a material (MoC_x) that convolutes traditional site-titration methods because of its own ability to bind common probe molecules like CO and H₂. However, there are a number of reasons that prevent me from recommending this manuscript for publication in Nature Communications. As noted, I'm not sure that this different (but not simpler or more readily accessible) method for quantifying Pt sites on a carbide catalyst significantly changes the understanding of these materials as the authors claim. I think there are major issues with the clarity of the figures, discussion, and language. The paper has many coined phrases that are undefined and, while I generally don't comment on grammar, the volume of typographical errors made it difficult to follow the logical arguments presented. There is also insufficient description of the catalyst synthesis, which in the case of molybdenum carbide is particularly worrisome as this catalyst is highly dynamic under atmospheric conditions and reaction conditions. For example, the premise of this paper relies on the pristine facets of MoC_{1-x} remaining intact as this is central to the modeling and geometric assumptions, yet the carbide catalyst is passivated (forming an oxide surface) and handled in air. No discussion at all is given to the potential for oxycarbide formation at all despite this being a prevailing theme in carbide catalysis. A few specific notes to make improvements to the paper are suggested below.

- The title is somewhat misleading as no tuning (or mention of tuning beyond the title and conclusion) is described.
- Sacrificial CO adsorption per Pt atom is written many places yet what this means is not intuitive or defined
- Grammar in the introduction is hard to follow with phrases like "obvious electronic interactions" throughout that are ambiguous.
- How was metal loading determined? This seems to be a pretty big omission if the actual metal loading wt% is not determined and instead relies on nominal loadings from the synthesis. Further, the synthesis details lack any reported values, so the reader can't assess if any errors have been made in the assumptions about loading.
- Figure one has several typos and is generally difficult follow. It is unclear what the colors represent or what kind of action "reasonable" is
- A general description of what is meant by "models" in the first paragraph of the Results section would help in understanding the figure and subsequent discussion.
- "Loading Rules" "Sacrificial CO adsorption per Pt atom" "CO adsorption rules" are all mentioned but not described or defined
- What is the small peak or shoulder between the 111 and 200 planes in present in the XRD patterns for most of the catalysts? Is there still bulk oxide present and detectable?
- What is "loss amounts of CO adsorption per Pt atom"
- No description of what is being shown in the inlay for the high mag STEM images nor what is circled.
- Are no Pt atoms or clusters visible in the STEM imaging? Surely some would be visible based on the modeled size of these in subsequent parts?
- The bamboo raft and presence of single layers of Pt are described, but where are they in the images?
- Why were Mo⁶⁺ and Mo⁴⁺ peaks omitted from the XPS fit. There is significant overlap between carbidic peaks and high oxidation state peaks associated with Mo-O as is widely reported in the literature that are very close to the 2 peaks that were fit. These peaks would be expected, particularly if the sample was passivated in 1% O₂ as the penetration depth of the XPS is limited.
- Why were DRIFTS experiments not conducted to evaluate the difference between Pt and Mo-based CO binding? This would be a fairly straightforward way to determine differences in binding sites no?
- What is "the law of CO adsorption"?

Version 1:

Reviewer comments:

Reviewer #2

(Remarks to the Author)

The response of the authors to my previous comments is satisfactory. They have substantially improved the quality of the article and, therefore, I recommend acceptance for publication.

Reviewer #3

(Remarks to the Author)

The authors have suitably addressed my suggestions. If this is also the case for the other reviewers, whose comments are outside of my area of expertise, I am comfortable with publication.

Responses to Editor's and Reviewers' Comments

Dear Editor,

Thank you very much for giving us the opportunity to revise the manuscript. We have revised the manuscript by fully taking into account the reviewers' comments and added more evidence to clarify the questions raised by reviewers. We hope that fits the requirements for publication.

Reviewer #1:

This manuscript highlights the high performance of Pt/MoC for low-temperature water gas shift reaction, which have been found previously. The quantitatively describing and of the active sites at the interface perimeter between Pt and MoC was concentrated on in order to clarify the underlying site configuration. The authors made significant endeavours to recognize the CO adsorption site and correlated it with active sites. However, there are many shortcomings that need further clarification.

(1) It is well known that with the increasing of the Pt loading amount, the metal size also evolved from single atoms to clusters and then nanoparticles. Firstly, the adsorption mode of CO can also change from linear mode to bridged one. Thus, how to precisely quantify the active sites from CO adsorption amount. Moreover, for a high loading amount of Pt, there should be a mixture of Pt single atoms and other clusters. It also affects the analysis of the contribution to CO adsorption.

Response:

We fully agree with your comments. As the Pt loading increases, Pt transitions from single atoms to single atomic layers, clusters, and finally to particles. CO molecules are adsorbed in bridge or tri-coordinate configurations on Pt clusters and particles, which affects the analysis of Pt cluster size by CO adsorption. The proposed method is more suitable for estimating the number of active sites on single sites and in monolayer configurations. CO molecules adsorb linearly on Pt and Mo atoms in a monolayer configuration. Due to the strong interaction between Pt and Mo carbide, Pt self-assembles as a monolayer at relatively low Pt loadings. This allows us to precisely measure the active sites and optimize the low-temperature WGS by optimizing the

monolayer cluster size, aligning with the method proposed in this paper. Further research is needed to calculate the active sites of Pt particle-sized catalysts. We have supplemented the scope of the application of this method in the abstract and discussion to address the issue:

Pt/ α -MoC_{1-x} catalysts exhibit exceptional activity in low-temperature water-gas shift (LTWGS) reactions. However, quantitatively identifying and fine-tuning the active sites has remained a significant challenge. In this study, we reveal that fully exposed monolayer Pt nanoclusters on molybdenum carbides demonstrate mass activity that exceeds that of bulk molybdenum carbide catalysts by one to two orders of magnitude at 100-200 °C for LTWGS. This advancement is driven by the precise quantification and elucidation of active sites along the Pt–molybdenum carbide interfacial perimeter. By combining sacrificial CO adsorption per Pt atom, Density Functional Theory (DFT) calculations, and CO chemisorption measurements, we establish a direct correlation between the monolayer Pt nanocluster size and the number of interfacial perimeters on Pt/ α -MoC_{1-x} catalysts. These findings provide key insights into the active site configuration of Pt/ α -MoC_{1-x} catalysts and open new pathways for innovative catalyst design, with the interfacial perimeter identified as a crucial factor in enhancing catalytic performance. (Manuscript, Page 2, line 11-13)

This research has significantly advanced our understanding of Pt/ α -MoC_{1-x} catalysts, particularly their remarkable efficacy in low-temperature water-gas shift reactions. Despite the inherent challenges in quantitatively characterizing and fine-tuning the active sites of these catalysts, our findings offer insights and a method for the measurement of active site numbers. We discovered that the mass activity of fully exposed monolayer Pt nanoclusters on molybdenum carbides is markedly higher than that of isolated single Pt atoms and the bulk molybdenum carbide catalysts, by an impressive one and two orders of magnitude, respectively, for low-temperature WGS at 100-200 °C. The cornerstone of this achievement lies in the detailed elucidation and quantification of the active sites situated along the Pt-molybdenum carbide interfacial perimeter. By leveraging a synergistic approach that combines the sacrificial amount of CO adsorption per Pt atom, Density Functional Theory (DFT) calculations, and CO

chemisorption measurements, we established a direct link between the number of interfacial perimeters and the size of monolayer Pt nanoclusters. This correlation not only sheds light on the intricate active site configuration of Pt/ α -MoC_{1-x} catalysts but also paves the way for the design of advanced catalysts. Our research emphasizes the critical role of the interfacial perimeter of fully exposed monolayer clusters in enhancing catalytic performance, thereby offering a new paradigm for catalyst development in a wide array of chemical reactions. However, the proposed method is limited to monolayer dispersed catalyst systems. The quantification of particle size and active sites of multi-layer atomic clusters and metal particles still needs to be improved on the basis of this method. (Manuscript, Page 24, line 10-Page 25, line 9)

(2) The manuscript considers that the states of Pt and Mo sites are not changed with the loading amount and the Pt size, which is not consistent with conventional conclusion. Particularly, for single atoms, there is coordination between Pt and Mo or C species, while for clusters or particularly nanoparticles, the Pt-Pt bond should exist. The different coordination environment can affect the state of Pt and Mo.

Response:

We agree with the changes regarding the coordination environment conventionally changing. However, both the literature research presented in Figures R1-R3¹⁻³ and our experimental characterization data have shown that the XPS and XANES results indicate that the states of Mo and the loaded metal Pt do not change with variations in loading within a certain range (0.2% to 2.0% loading). relative stable oxidation state is likely related to the single-site and single atomic monolayer structure over a relatively large range of Pt loadings. To investigate the electronic structure around different cluster sizes, we constructed Pt₁/ α -MoC_{1-x}, Pt₄/ α -MoC_{1-x}, Pt₈/ α -MoC_{1-x}, and three kinds of Pt₂₄/ α -MoC_{1-x}-cluster models by DFT, which include single-layer, double-layer, and three-layer configurations. As shown in Fig. S10, the Bader charges of atoms were calculated, and the Bader charge of Mo atoms at the interface sites remained between -0.70 eV and -0.76 eV. The difference in charge values is relatively small, indicating that the intrinsic activity of interface sites is indistinguishable.

In the present work, Pt is mostly distributed automatically in a monolayer on Mo carbides within the studied range of Pt loadings. The difference in electronic properties between single-site and single atomic monolayers seems to be indistinguishable within the experimental limitations. The content is supplemented on page 10 as follows:

Moreover, Bader charge analysis of Mo atoms at the interface sites across different models ($\text{Pt}_1/\alpha\text{-MoC}_{1-x}$, $\text{Pt}_4/\alpha\text{-MoC}_{1-x}$, $\text{Pt}_8/\alpha\text{-MoC}_{1-x}$, and three variants of $\text{Pt}_{24}/\alpha\text{-MoC}_{1-x}$ with single, double, and triple layers) revealed small charge variations (-0.70 to -0.76 eV), indicating minimal differences in the intrinsic activity of the interface sites (Fig. S10). (Manuscript, Page 9, line 11-15)

The sentence “In summary, the experimental and DFT calculations demonstrated that Mo active sites and Pt atoms have similar chemical environments over $\text{Pt}/\alpha\text{-MoC}_{1-x}$ catalysts with different loading amounts.” is modified to:

Both experimental results and DFT calculations demonstrate that Mo active sites and Pt atoms exhibit similar chemical environments across $\text{Pt}/\alpha\text{-MoC}_{1-x}$ catalysts with relatively low Pt loadings. (Manuscript, Page 9, line 15-17)

Fig. S10 $\text{Pt}_n/\alpha\text{-MoC}-111$ model (a) Pt_1 , (b) Pt_4 , (c) Pt_4 , (d) Pt_{24} (monolayer), (e) Pt_{24} (bilayer), and (f) Pt_{24} (bilayer). The numerical value represents the charge of the corresponding atom. Pt(blue), Mo(green color), C(grey color).

Fig. R1 Pt L3-edge XANES spectra of Pt/ α -MoC_{1-x} catalysts with different Pt loadings¹

Fig. R2 Pt L3-edge XANES spectra for the Pt/ α -MoC_{1-x} catalysts²

Fig. R3. Structure characterization of the as-prepared Ni/ α -MoC catalysts. (a) XRD profiles of the Ni/ α -MoC with different Ni loadings. (b) In situ XPS of 0.5%, 2%, and 7% Ni/ α -MoC catalysts. The Ni 2p, C 1s, and Mo 3d regions are presented. (c) Ni K edge XANES spectra of 0.5%, 2%, 3%, 5%, and 7% Ni/ α -MoC catalysts along with Ni foil and Ni oxide. The enlarged XANES white-line and pre-edge region are shown in the right panel. (d) Ni K edge EXAFS spectra and fittings of 0.5%, 2%, 3%, 5%, and 7% Ni/ α -MoC catalysts³.

(3) It is reasonable to study the location of Pt species on the face of MoC, particularly it has been found that there is (111) and (100) facet. However, the evidence of the location state is not ample. Which one is more stable, on MoC (111) or (100). Moreover, it needs more evidence to identify the location sites of MoC.

Response:

Fig. 1 Textural properties of the Pt/ α -MoC_{1-x} catalysts with varying Pt loadings: 0.2%, 0.4%, 1.0% and 2.0%. (a) XRD diffraction patterns. (b-c) N₂ adsorption-desorption isotherms, and (c) pore size distribution profiles for the four catalysts. (d1-d3) AC-HAADF-STEM images of 0.2% Pt/ α -MoC_{1-x} catalyst. (d4) Line-profile intensities of Pt atomic layers corresponding to panel (d1). (e1-e3) AC-HAADF-STEM image of 1.0% Pt/ α -MoC_{1-x} catalyst. (e4) Line-profile intensities of Pt atomic layers corresponding to panel (e3). Pt₁ species are highlighted with green dashed circles, and Pt clusters (Pt_n) are highlighted with red ellipses.

Fig. S5 HRTEM images of the Pt/ α -MoC_{1-x} catalysts: (a) 0.2% Pt/ α -MoC_{1-x}; (b) 1.0% Pt/ α -MoC_{1-x}; (c) 2.0% Pt/ α -MoC_{1-x}.

Table R1 The binding energies between Pt single atoms and different crystal planes

	E_{surface} (eV)	$E_{\text{Pt/surface}}$ (eV)	binding energy(eV)
α -MoC-100	-702.69	-708.09	0.67
α -MoC-111	-615.58	-622.27	-0.63

Thank you very much for your suggestion. We have added more experimental results to show the Pt location. As shown in Fig. S5, HRTEM images show that α -MoC_{1-x} exposes (100) and (111) facets. As shown in Fig. 1, AC-HAADF-STEM images of Pt/ α -MoC_{1-x} catalysts show that Pt atoms are observed on both (111) and (100) facets. It has not been possible to determine in which crystal plane Pt atoms are more stable through experimental characterization methods. In addition, the interaction between Pt atoms and different crystal planes was studied by DFT. As shown in Table R1, it was found that the binding energy of Pt with the (111) facet was higher than that with the (100) facet, meaning more stable on (111). Based on the conclusions from the experimental characterization, the Pt_n/ α -MoC-111 and Pt_n/ α -MoC-100 structures were determined using DFT calculations. In order to better improve the quality of manuscript, we have modified as follows:

Further analysis using high-resolution transmission electron microscopy (HR-TEM) and aberration-corrected high-angle annular dark-field scanning transmission electron microscopy (AC-HAADF-STEM) (Fig. S5 and 1(d1-e4)) showed that the exposed crystal planes of the α -MoC_{1-x} support are the (111) and (200) planes^{4,6,15,27,28}. The Pt species were uniformly dispersed, with no visible particles. (Manuscript, Page 7, line 10-14).

(4) MoC is recognized to be a noble-metal like support that means it can easily adsorb CO or H₂ like noble-metal species. The method of sacrificial CO adsorption per Pt atom may be effective to distinguish the adsorption on Pt and MoC sites. Are there any CO spillover effects that can influence the quantification? If there is no spillover effect, the use of H₂ chemisorption or H₂-O₂ titration experiment can further support these results.

Response:

Thank you very much for your suggestion. The spillover phenomenon refers to the phenomenon that the active centre (original active centre) on the surface of the solid catalyst produces an ionic or free radical active species through adsorption, and they migrate to other active centres (secondary active centres). They can induce new activity or carry out certain chemical reactions through chemical adsorption. Firstly, the DFT calculations and previous literature research conclusions show that CO molecules are adsorbed on the surfaces of both Pt metal and α -MoC_{1-x} support (Fig. S3), which suppress CO spillover effects on the Pt/catalyst surface.

Secondly, the adsorption behaviour of CO and H at different sites was analysed by DFT calculation. As shown in Fig. S3 and Table S1, the CO adsorption configurations on the α -MoC-111, α -MoC-100, Pt/ α -MoC-111, Pt/ α -MoC-100, and Pt-111 are all linear and adsorption energies at different sites are slightly different. Both Pt and Mo sites have strong adsorption strength for CO molecules. As shown in Fig. S2 and Table S1, there are two configurations of H adsorption on different surfaces: three-coordinate adsorption and linear adsorption. It is worth noting that the adsorption energy of H at the Pt site on the Pt₁/ α -MoC-111 model and the Pt₁/ α -MoC-100 model is positive, which means that the adsorption strength is weak. This variation in H adsorption configuration makes it challenging to accurately quantify Pt and Mo sites. Therefore, calculating the cluster size of Pt by the difference in the adsorption amount of CO on different catalysts is the most feasible solution.

Third, the H₂-O₂ titration experiment is also not feasible. The reason is that α -MoC_{1-x} is highly sensitive in the presence of oxygen and the surface Mo carbide will spontaneously oxidize, making it difficult to quantify the adsorbed H by H₂-O₂ titration.

Therefore, we chose to quantitatively analyse the Pt cluster size by CO chemical adsorption. The reason why was CO molecule chosen as the probe molecule for quantitative analysis of cluster size and active sites are supplemented in the Electronic Supplementary Material. As follows:

(1) Why was the CO molecule chosen as the probe molecule for quantitative analysis of cluster size and active sites?

The spillover phenomenon refers to the phenomenon that the active centre (original active centre) on the surface of the solid catalyst produces an ionic or free radical active species through adsorption, and they migrate to other active centres (secondary active centres). They can induce new activity or carry out certain chemical reactions through chemical adsorption. Firstly, the DFT calculations and previous literature research conclusions show that CO molecules are adsorbed on the surfaces of both Pt metal and α -MoC_{1-x} support (Fig. S3), which suppress CO spillover effects on the Pt/catalyst surface.

Secondly, the adsorption behaviour of CO and H at different sites was analysed by DFT calculation. As shown in Fig. S3 and Table S1, the CO adsorption configurations on the α -MoC-111, α -MoC-100, Pt/ α -MoC-111, Pt/ α -MoC-100, and Pt-111 are all linear, and adsorption energies at different sites are slightly different. Both Pt and Mo sites have strong adsorption strength for CO molecules. As shown in Fig. S2 and Table S1, there are two configurations of H adsorption on different surfaces: three-coordinate adsorption and linear adsorption. It is worth noting that the adsorption energy of H at the Pt site on the Pt₁/ α -MoC-111 model and the Pt₁/ α -MoC-100 model is positive, which means that the adsorption strength is weak. This variation in H adsorption configuration makes it challenging to accurately quantify Pt and Mo sites. Therefore, calculating the cluster size of Pt by the difference in the adsorption amount of CO on different catalysts is the most feasible solution.

Third, the H₂-O₂ titration experiment is also not feasible. The reason is that α -MoC_{1-x} is highly sensitive in the presence of oxygen and the surface Mo carbide will spontaneously oxidize, making it difficult to quantify the adsorbed H by H₂-O₂ titration.

Therefore, we chose to quantitatively analyse the Pt cluster size by CO chemical adsorption. (Electronic Supplementary materials, supplementary note 2, Page 3, line 1-28)

Fig. S2 Adsorption configurations of H atom on three models. (a) α -MoC-111, (b) Pt-111, (c) Pt₁/ α -MoC-111, (d) Pt₁/ α -MoC-100. Pt(blue colour), Mo(green colour), C(grey colour), H(white colour).

Fig. S3 Adsorption configurations of CO molecule on three models. (a) α -MoC-111, (b) Pt-111, (c) Pt₁/ α -MoC-111, (d) Pt₁/ α -MoC-100. Pt(blue colour), Mo(green colour), C(grey colour), O(red colour).

Table S1 Adsorption energies of H species on different models

Adsorbate	α -MoC-111			Pt-111			Pt ₁ / α -MoC-111			Pt ₁ / α -MoC-100		
	No.	Site	E _{ads} (eV)	No.	Site	E _{ads} (eV)	No.	Site	E _{ads} (eV)	No.	Site	E _{ads} (eV)
H	1	top-Mo	-0.43	1	fcc-Pt	-0.44	1	top-Mo	-0.25	1	top-Pt	-0.35
	2	fcc-Mo	-0.50	2	fcc-Pt	-0.41	2	fcc-Mo	-0.26	2	top-Mo	0.22
	3	fcc-Mo	-0.74	3	top-Pt	-0.44	3	top-Mo	-0.37	3	top-C	-0.38
							4	fcc-Mo	-0.79			
							5	top-Pt	0.03			
CO	1	O-top-Mo	-0.06	1	fcc-Pt	-1.77	1	C-top-Pt	-1.30	1	C-top-Mo	-1.23
	2	C-top-Mo	-2.00	2	--	-0.004	2	C-top-Mo(Pt)	-1.68	2	C-top-Pt	-2.35
	3	--	3.82	3	top-Pt	-1.59	3	O-top-Mo	0.13	3	C-top-Pt	-2.31
	4	C-top-Mo	-1.98	4	--	-0.01	4	C-top-Mo	-1.96			
	5	--	-0.001	5	fcc-Pt	-1.71	5	--	-0.01			
	6	C-top-Mo	-1.98				6	C-top-Pt	-1.30			
						7	C-top-Pt	-1.30				
						8	C-top-Mo	-2.03				

(5) The DFT calculation can exactly study the effects of Pt layers while the experimental evidences are not enough. The AC-HAADF STEM may provide more information, which need more characterization from different perspectives.

Response:

Thank you very much for your suggestion. We have added more experimental results. To characterize the accurate morphology of Pt, we determined the atomic layer

thickness of Pt on different catalysts by comparing the atomic layer thickness of Pt with the line profile intensity of isolated atoms. As shown in Fig. 1(d4), Fig. 1(e4) and Fig. S7-S8, it is proved that it is shown that Pt atoms are mainly dispersed in a single layer on 0.2%Pt/ α -MoC_{1-x} and 1%Pt/ α -MoC_{1-x} catalysts. The manuscript was modified on page 8 as follows:

On the 0.2% Pt/ α -MoC_{1-x} catalyst, Pt primarily exists as single atoms and small clusters (Fig. 1(d1-d3)). The line profile intensity of the isolated atoms (Fig. 1(d4), Fig. 1(e4), and Figs. S7-S8) corresponds to Pt atom sizes, indicating that the small clusters form an atomic monolayer on the α -MoC_{1-x} surface, rather than being embedded within the α -MoC_{1-x} structure (details in Supplementary Note 1). In the 1.0% Pt/ α -MoC_{1-x} catalyst, Pt also shows a monolayer distribution, resembling a “bamboo raft” arrangement, but with significantly larger clusters than in the 0.2% Pt/ α -MoC_{1-x} sample.

(Manuscript, Page 7, line 14-21)

Fig. 1 Textural properties of the Pt/ α -MoC_{1-x} catalysts with varying Pt loadings: 0.2%, 0.4%, 1.0% and 2.0%. (a) XRD diffraction patterns. (b-c) N₂ adsorption-desorption isotherms, and (c) pore size distribution profiles for the four catalysts. (d1-d3) AC-HAADF-STEM images of 0.2% Pt/ α -MoC_{1-x} catalyst. (d4) Line-profile intensities of Pt atomic layers corresponding to panel (d1). (e1-e3) AC-HAADF-STEM image of 1.0% Pt/ α -MoC_{1-x} catalyst. (e4) Line-profile intensities of Pt atomic layers

corresponding to panel (e3). Pt₁ species are highlighted with green dashed circles, and Pt clusters (Pt_n) are highlighted with red ellipses.

Fig. S7 The line-profile intensities of Pt atomic layers of 0.2% Pt/ α -MoC_{1-x} sample.

Fig. S8 The line-profile intensities of Pt atomic layers of 1.0% Pt/ α -MoC_{1-x} sample.

Reviewer #2:

Very interesting study in a hot topic these days. The following points must be addressed by the authors:

Response:

(1) Bottom of page 7: Could the Pt be embedded in the carbide substrate?

Response:

Fig. S20 The Pt₁/α-MoC-111-C_{vac}, Pt₁/α-MoC-111-Mo_{vac}, Pt₁/α-MoC-111, Pt₁/α-MoC-100-C_{vac}, Pt₁/α-MoC-100-Mo_{vac}, Pt₁/α-MoC-100 models and coordination numbers of Pt-Mo bond or Pt-C bond. Pt (blue colour), Mo (green colour), C (grey colour).

Firstly, the thickness of the Pt atomic layer can be determined by comparing the thickness of the Pt atomic layer with the line profile intensity of the isolated atoms (Fig. 1(d4), Fig. 1(e4), and Fig.S7-S8). The Pt atoms are dispersed in a single atomic layer on the α-MoC_{1-x} surface.

Secondly, the coordination numbers of Pt with C and Mo on Pt₁/α-MoC-111-C_{vac}, Pt₁/α-MoC-111-Mo_{vac}, Pt₁/α-MoC-111, Pt₁/α-MoC-100-C_{vac}, Pt₁/α-MoC-100-Mo_{vac},

Pt₁/α-MoC-100 models were counted (as shown in Fig. S20). The results showed that the coordination number of Pt embedded in carbide substrate increased significantly. On the 0.2% Pt/α-MoC_{1-x} catalyst with single-atom dispersion, the coordination numbers of Pt-C and Pt-Mo are lower than those of the embedded model (Pt₁/α-MoC-111-C_{vac}, Pt₁/α-MoC-111-Mo_{vac}, and Pt₁/α-MoC-100-Mo_{vac}).

In summary, we infer that Pt atom is not embedded in the carbide substrate.

And Pt atom is not considered to be embedded in the carbide substrate (detailed explanation is given in the electronic supplementary material in the supplementary note 1 as follows:

Supplementary note 1

Could the Pt be embedded in the carbide substrate?

Firstly, the thickness of the Pt atomic layer can be determined by comparing the thickness of the Pt atomic layer with the line profile intensity of the isolated atoms (Fig. 1(d4), Fig. 1(e4), and Fig.S7-S8). The Pt atoms are dispersed in a single atomic layer on the α-MoC_{1-x} surface.

Secondly, the coordination numbers of Pt with C and Mo on Pt₁/α-MoC-111-C_{vac}, Pt₁/α-MoC-111-Mo_{vac}, Pt₁/α-MoC-111, Pt₁/α-MoC-100-C_{vac}, Pt₁/α-MoC-100-Mo_{vac}, Pt₁/α-MoC-100 models were counted (as shown in Fig. S20). The results showed that the coordination number of Pt embedded in carbide substrate increased significantly. On the 0.2% Pt/α-MoC_{1-x} catalyst with single-atom dispersion, the coordination numbers of Pt-C and Pt-Mo are lower than those of the embedded model (Pt₁/α-MoC-111-C_{vac}, Pt₁/α-MoC-111-Mo_{vac}, and Pt₁/α-MoC-100-Mo_{vac}).

In summary, we infer that Pt atom is not embedded in the carbide substrate.

(Electronic Supplementary Material, Page 2, line 1-14)

(2) In figure 3c, how stable is the species with C-C bonds?

Response:

The C-C bond is the peak of organic carbon contamination during XPS testing, and the content of C-C on the catalyst surface is relatively low. In our previous work (Chemical Engineering Journal 474 (2023) 145645), we demonstrated that by controlling the carbonization flow rates, it was found that there was no significant

carbon enrichment on the surface of the preparation conditions in this work (1g Pt/MoO₃ (40–60 mesh) was placed in a fixed bed reactor, 15% CH₄/H₂ mixed gas (50 mL·g⁻¹·min⁻¹) was introduced, the temperature was raised to 700 °C at 5 °C·min⁻¹, and kept for 2 h).

(3) Page 11, can the authors rule out adsorption of CO on the Pt-MoC_{1-x} interface?

Response:

Thank you very much for your suggestion. Regarding the issue of whether CO molecules adsorb at the Pt-MoC_{1-x} interface, the adsorption energies of two adsorption configurations were compared. As shown in Fig. S14, the adsorption configurations of four CO molecules at Pt-MoC_{1-x} interface and non-Pt-MoC_{1-x} interface sites were optimized. The results show that the CO molecules migrate from the Pt-MoC_{1-x} interface sites, and preferred to adsorb on the adjunct sites of the interface. The adsorption configuration at non-Pt-MoC_{1-x} interface sites can stably adsorb onto Mo sites. Moreover, comparing the adsorption energy, it was found that the non-Pt-MoC_{1-x} interface adsorption configuration is more stable than the Pt-MoC_{1-x} interface adsorption configuration. Therefore, we supplemented this set of data in the “CO adsorption rules” section to provide more comprehensive evidence for the CO adsorption rules. The manuscript was modified on page 16 as follows:

The adsorption behaviour of CO molecules at Pt and Mo sites was systematically studied to establish the “CO adsorption rules” on Pt/ α -MoC_{1-x} surfaces. These rules define the preferred adsorption sites and configurations of CO molecules, which vary based on the catalyst type. As shown in Fig. S14(a), on the α -MoC-111 surface, CO molecules adopt the O=C···Mo configuration, with an adsorption energy of -1.68 eV. When two CO molecules are adsorbed on the α -MoC-111 model, each CO molecule binds independently to adjacent Mo sites, resulting in identical adsorption configurations without repulsion between the CO molecules. In the Pt₄/ α -MoC-111 model, the mono-coordinated CO adsorption configuration at the Pt site is more stable than the tri-coordinated one. On the α -MoC-100 surface, two CO molecules can adsorb linearly on adjacent Mo sites, as depicted in Fig. S14(b). Although the initial configuration considered multi-coordinated CO adsorption on Pt, optimization revealed

that CO prefers to adsorb in a linear fashion. As shown in Fig. S15, the adsorption configurations of CO molecules at both Pt- α -MoC_{1-x} interface and non-Pt- α -MoC_{1-x} interface sites were optimized. The adsorption energies on the non-Pt- α -MoC_{1-x} interface configuration is more negative than that on the Pt- α -MoC_{1-x} interface configuration. The results indicate that CO molecules migrate from Pt- α -MoC_{1-x} interface sites and preferentially adsorb on adjacent sites of the interface. Meanwhile, at non-Pt- α -MoC_{1-x} interface sites, CO molecules stably adsorb onto Mo sites. This conclusion is also supported by the adsorption energy comparison at different sites, as shown in Fig. S13.

Based on the DFT and experimental results, CO adsorption rule was established, which refers to the guidelines governing the adsorption sites and configurations of CO molecules on Pt/MoC_{1-x} surfaces. The CO adsorption rules on Pt_n/ α -MoC-111 and Pt_n/ α -MoC-100 structures can be summarized as follows: (1) CO molecules adsorb linearly on adjacent Mo sites, with each Mo site binding to one CO molecule; (2) CO molecules also adsorb linearly on Pt sites, with each exposed Pt atom binding to one CO molecule; and (3) Mo atoms coordinated with Pt lose the ability to adsorb CO molecules. (Manuscript, Page 15, line 20-Page 17, line 4)

Fig. S13 Adsorption configurations of CO molecular on Pt₁/α-MoC-111 model.

Pt(blue colour), Mo(green colour), C(grey colour).

Fig. S15 The adsorption configuration of CO molecules at Pt-MoC_{1-x} interface and non Pt-MoC_{1-x} interface sites. Pt (blue colour), Mo (green colour), C (grey colour).

(4) Top of page 13, the authors need to give a better justification of the models used in the DFT calculations.

Response:

First, it was found that the main exposed crystal faces of the α -MoC_{1-x} were (111) and (200) by XRD and AC-HAADF-STEM characterization analysis. Secondly, the AC-HAADF-STEM characterization results showed that the dispersion state of Pt was single atom or single atomic layer when the Pt loading was relatively low. As the loading increased, the size of single-layer Pt clusters increased significantly, and multi-atomic layers appeared. Therefore, the α -MoC-111 and α -MoC-100 crystal planes were selected as support models in the model building part. The binding rules with Pt on different crystal planes were studied by DFT calculation (as shown in Fig. S11-Fig. S12). Pt_n/ α -MoC-100 and Pt_n/ α -MoC-111 models with different Pt cluster sizes were built according to the binding rules. The models and experimental results were linked by the “sacrificial amount of CO adsorption per Pt atom” parameter. As shown in Fig. 5(c), the results show that the Pt structure inferred by the DFT calculation is consistent with the results of EXAFS and AC-HAADF-STEM characterization, which means that the construction of the DFT model is reasonable.

(5) Pages 22-23, the "Discussion" is kind of short and looks like a "Conclusions" section. The article will benefit from a larger and more polished Discussion.

Response:

Thank you very much for your suggestion. In Nature Communications, the discussion section is more like a conclusion section. Anyhow, we made the following additions to the "Discussion" section.

In supported catalysts, the complex and varied surface and interface structures make it difficult to identify and quantify active sites. The quantification of active sites is often closely related to the particle size of the supported metal. On Pt/ α -MoC_{1-x} catalysts, many studies have demonstrated that Pt atoms are dispersed on the α -MoC_{1-x} surface in a form of monolayer due to the strong metal-support interaction of Pt- α -MoC_{1-x}^{1, 12, 13, 17}. It is very difficult to count the cluster size of Pt on the Pt/ α -MoC_{1-x} catalysts. A combined DFT calculation and chemisorption method is proposed to quantify Pt cluster size and the number of active sites.

This research has significantly advanced our understanding of Pt/ α -MoC_{1-x} catalysts, particularly their remarkable efficacy in low-temperature water-gas shift reactions. Despite the inherent challenges in quantitatively characterizing and fine-tuning the active sites of these catalysts, our findings offer insights and a method for the measurement of active site numbers. We discovered that the mass activity of fully exposed monolayer Pt nanoclusters on molybdenum carbides is markedly higher than that of isolated single Pt atoms and the bulk molybdenum carbide catalysts, by an impressive one and two orders of magnitude, respectively, for low-temperature WGS at 100-200 °C. The cornerstone of this achievement lies in the detailed elucidation and quantification of the active sites situated along the Pt-molybdenum carbide interfacial perimeter. By leveraging a synergistic approach that combines the sacrificial amount of CO adsorption per Pt atom, Density Functional Theory (DFT) calculations, and CO chemisorption measurements, we established a direct link between the number of

interfacial perimeters and the size of monolayer Pt nanoclusters. This correlation not only sheds light on the intricate active site configuration of Pt/ α -MoC_{1-x} catalysts but also paves the way for the design of advanced catalysts. Our research emphasizes the critical role of the interfacial perimeter of fully exposed monolayer clusters in enhancing catalytic performance, thereby offering a new paradigm for catalyst development in a wide array of chemical reactions. However, the proposed method is limited to monolayer dispersed catalyst systems. The quantification of particle size and active sites of multi-layer atomic clusters and metal particles still needs to be improved on the basis of this method. (Manuscript, Page 24, line 1-Page 25, line 9)

Reviewer #3:

The authors present the synthesis of a series of Pt/MoC_{1-x} with varying Pt loadings and use this series to evaluate a model for particle/interface size and correlate this to catalytic activity for the water gas shift reaction. I think this is an interesting approach, but struggle to see the same novelty as the authors. The paper attempts to address the challenges associated with characterizing Pt-based sites on a material (MoC_x) that convolutes traditional site-titration methods because of its own ability to bind common probe molecules like CO and H₂. However, there are a number of reasons that prevent me from recommending this manuscript for publication in Nature Communications. As noted, I'm not sure that this different (but not simpler or more readily accessible) method for quantifying Pt sites on a carbide catalyst significantly changes the understanding of these materials as the authors claim. I think there are major issues with the clarity of the figures, discussion, and language. The paper has many coined phrases that are undefined and, while I generally don't comment on grammar, the volume of typographical errors made it difficult to follow the logical arguments presented. There is also insufficient description of the catalyst synthesis, which in the case of molybdenum carbide is particularly worrisome as this catalyst is highly dynamic under atmospheric conditions and reaction conditions. For example, the premise of this paper relies on the pristine facets of MoC_{1-x} remaining intact as this is central to the modeling and geometric assumptions, yet the carbide catalyst is passivated (forming an oxide surface) and handled in air. No discussion at all is given to the potential for oxycarbide formation at all despite this being a prevailing theme in carbide catalysis. A few specific notes to make improvements to the paper are suggested below.

Thank you for your positive feedback on the innovative approach. However, quantifying the interfacial site number remains a significant challenge. Traditional methods like CO and H₂ chemisorption are unable to accurately determine the interface number. Our developed method addresses this by enabling the determination of the interfacial site number, identifying the active sites, and optimizing the size of the atomic layer cluster to achieve the maximum reaction rate. We apologize for the use of many

coined phrases in the manuscript, which may have made evaluation difficult. We have revised the manuscript to improve its clarity and readability. This version aims for a more polished and concise flow while retaining the original meaning.

(1) The title is somewhat misleading as no tuning (or mention of tuning beyond the title and conclusion) is described.

Response:

The title was modified to “Quantification and optimization of platinum–molybdenum carbide interfacial sites to enhance low-temperature water-gas shift reaction”. The title highlights the key findings of this work, which include: (1) the first-ever determination of the interfacial site number, (2) the identification of platinum–molybdenum carbide interfacial sites as the active catalytic sites, (3) the discovery of self-assembly of Pt into an atomic layer on α -MoC_{1-x}, and (4) the optimization of active site density by increasing the size of the atomic layer cluster to enhance the water-gas shift (WGS) reaction

(2) Sacrificial CO adsorption per Pt atom is written many places yet what this means is not intuitive or defined.

Response:

Firstly, we apologize for the issue of using multiple words to describe the same meaning in the manuscript. “sacrificial CO adsorption per Pt atom”, “sacrificial amount of CO adsorption per Pt atom”, and “loss amounts of CO adsorption per Pt atom” all represent the reduction amount of CO molecules adsorbed on the α -MoC_{1-x} surface after loading a Pt atom on the α -MoC_{1-x} surface. We are very sorry that this issue caused great difficulties for readers. The entire manuscript has been uniformly revised and the “sacrificial amount of CO adsorption per Pt atom” was chosen.

Secondly, thank you for your suggestion. In the Fig S1, the definition of the “Sacrificial amount of CO adsorption per Pt atom” parameter has been added. As shown below:

Fig. S1 The research concept diagram for determining Pt_n nanocluster size through CO adsorption. The “Sacrificial amount of CO adsorption per Pt atom” parameter is defined as the reduction amount of CO molecules adsorbed on the α - MoC_{1-x} surface after loading a Pt atom on the α - MoC_{1-x} surface. (Electronic Supplementary Material, Page 3, line 13-16)

(3) Grammar in the introduction is hard to follow with phrases like “obvious electronic interactions” throughout that are ambiguous.

Response:

Thank you very much for your suggestion. The professional teacher of our team has made modifications to grammatical errors and ambiguous expressions.

(4) How was metal loading determined? This seems to be a pretty big omission if the actual metal loading wt% is not determined and instead relies on nominal loadings from the synthesis. Further, the synthesis details lack any reported values, so the reader can’t assess if any errors have been made in the assumptions about loading.

Response:

We agree that metal loading is a critical parameter in our method. Thank you very much for your valuable suggestions for improving paper. First, the Pt contents of 0.2% Pt/α - MoC_{1-x} , 0.4% Pt/α - MoC_{1-x} , 1.0% Pt/α - MoC_{1-x} , and 2.0% Pt/α - MoC_{1-x} catalysts were measured by inductively coupled plasma atomic emission spectroscopy (ICP-AES). In order to ensure that the samples were not oxidized and the content was accurately measured, a certain amount of aqua regia was used to digest the samples in the glove box before ICP determination. The results showed that the Pt contents of the

0.2% Pt/ α -MoC_{1-x}, 0.4% Pt/ α -MoC_{1-x}, 1.0% Pt/ α -MoC_{1-x}, and 2.0% Pt/ α -MoC_{1-x} catalysts were 0.22%, 0.41%, 1.02%, and 2.03%, respectively. The nanocluster size, number of Pt_n nanoclusters, and interface perimeter per gram of catalyst were recalculated using the exact Pt contents, and the conclusions were found to be consistent with those before the modification.

In addition, the preparation process of the catalysts and the ICP characterization method were supplemented in the methods section. As follows:

The actual contents of Pt were measured with inductively coupled plasma atomic emission spectroscopy (ICP-AES). In order to ensure that the samples were not oxidized and the content was accurately measured, a certain amount of aqua regia was used to digest the samples in the glove box before ICP determination. (Manuscript, Page 28, line 21- Page 29, line 3)

Table S2 Textural properties of the Pt/ α -MoC_{1-x} catalysts with different loadings

Catalyst	Cell parameters (a, b, c, α , β , γ) ^a	BET surface area(cm ³ ·g ⁻¹) ^b	Single point Pore volume(cm ³ ·g ⁻¹) ^b	BJH Desorption Pore size(nm) ^b	Pt content (wt%) ^c
0.2% Pt/ α - MoC _{1-x}	3.01,3.01,14.61 (90, 90, 120)	76.71	0.17	7.00	0.19
0.4% Pt/ α - MoC _{1-x}	2.99, 2.99, 14.52 (90, 90, 120)	84.41	0.17	6.21	0.41
1.0% Pt/ α - MoC _{1-x}	2.99, 2.99, 14.52 (90, 90, 120)	72.83	0.17	6.93	1.07
2.0% Pt/ α - MoC _{1-x}	2.99, 2.99, 14.52 (90, 90, 120)	71.11	0.16	7.08	2.10

^aDetermined by XRD. ^bDetermined by BET. ^cDetermined by ICP-AES.

(5) Figure one has several typos and is generally difficult follow. It is unclear what the colours represent or what kind of action “reasonable” is?

Response:

Fig. S1 The research concept diagram illustrates the determination of Pt_n nanocluster size through CO adsorption, using a combined DFT (blue) and experimental (red) approach. The parameter “sacrificial amount of CO adsorption per Pt atom” is defined as the decrease in the number of CO molecules adsorbed on the α - MoC_{1-x} surface following the deposition of a Pt atom on the α - MoC_{1-x} surface.

Thank you very much for your suggestion. Based on doubts about Fig. S1, we have made modifications to Fig. S1. As shown in Fig. S1. blue represents the model section, and red represents the experimental section. The term “reasonable” refers to judging whether the model selection is reasonable based on the relationship results. After careful consideration, the word “reasonable” is not reasonable, so it is removed. The reason is that the rationality of the catalyst model should be determined in conjunction with structural characterization.

(6) A general description of what is meant by “models” in the first paragraph of the Results section would help in understanding the figure and subsequent discussion.

Response:

Thank you very much for your suggestion. In the first paragraph of results, the “models” was supplemented and explained. As shown below:

As illustrated in Fig. S1, the research concept of this study is outlined. In the first step, a series of Pt_n/α - MoC -111 and Pt_n/α - MoC -100 models (representing the microstructures of Pt atoms on the α - MoC -111 and α - MoC -100 surfaces, respectively) with varying cluster sizes were constructed in DFT based on Pt loading rules. The “sacrificial amount of CO adsorption per Pt atom” parameters for different models were

determined according to the CO adsorption rules. Additionally, the “sacrificial amount of CO adsorption per Pt atom” parameters of the catalysts were experimentally determined using CO pulse experiments, establishing a meaningful link between the models and real catalysts. The microstructure models of catalysts with different loadings were further refined using EXAFS characterization. In the second step, the total number of active sites in the catalyst was calculated by identifying the active sites and counting Pt clusters. In the third step, the mass activities of catalysts with different loadings were correlated to the total number of active sites, establishing a benchmark for active site determination and providing a guide for optimizing catalyst structure to maximize the number of active sites. (Manuscript, Page 5, line 18- Page 6, line 10)

(7) “Loading Rules” “Sacrificial CO adsorption per Pt atom” “CO adsorption rules” are all mentioned but not described or defined

Response:

Thank you very much for your suggestion.

“Sacrificial CO adsorption per Pt atom”, also known as “sacrificial amount of CO adsorption per Pt atom”, and its definition was replied to in question (2).

“Loading Rules” refers to a rule that metal Pt combines with the surface of molybdenum carbide, including binding sites and forms, ultimately affecting the presentation state (single atom, single atomic layer, particles, clusters) of Pt atom on the surface. This explanation was added to the manuscript page 16. As follows:

Based on the experimental results, the loading rules are defined which refer to guidelines that describe how metal Pt interacts with the surface of molybdenum carbide, including the binding sites and configurations. (Manuscript, Page 15, line 9-11)

“CO adsorption rules” refer to a rule followed by the adsorption sites and configurations of CO molecules on Pt/MoC_{1-x} surfaces, and the adsorption rules will change with the type of catalyst. This explanation was added to the manuscript. As follows:

These rules define the preferred adsorption sites and configurations of CO molecules, which vary based on the catalyst type (Manuscript, Page 15, line 21- Page 16, line 1)

(8) What is the small peak or shoulder between the 111 and 200 planes in present in the XRD patterns for most of the catalysts? Is there still bulk oxide present and detectable?

Response:

After reviewing the literatures, the peak (39.5°) between (111) and (200) is assigned to β -MoC^{1, 4-6}. The oxidizing atmosphere only oxidizes a thin layer of the surface. This does not cause phase change or the appearance of large oxidized species. A supplementary description of XRD characterization is given in the “Geometric structure of Pt/ α -MoC_{1-x} catalysis” section. As follows:

As shown in Fig. 1(a), XRD characterization revealed four characteristic peaks of α -MoC_{1-x} crystals (PDF #08-0384) at 36.9° , 42.6° , 62.3° , and 74.5° ^{12,16,27}, alongside a peak at 39.5° corresponding to β -MoC^{12,13}. (Manuscript, Page 7, line 7-10)

(9) What is “loss amounts of CO adsorption per Pt atom”

Response:

Thank you very much for your reminder and suggestions.

“Loss amounts of CO adsorption per Pt atom” was modified to “Sacrificial amount of CO adsorption per Pt atom”, and its meaning is answered in both the question (2) and question (7).

(10) No description of what is being shown in the inlay for the high mag STEM images nor what is circled.

Response:

We are very sorry that the description of Fig. 1 is not sufficient and has caused inconvenience to read. In order to better interpret the information in the figure, supplementary explanations were provided in the geometric structure of Pt/ α -MoC_{1-x} catalogue. The supplementary content is as follows:

Fig. 1 Textural properties of the Pt/ α -MoC_{1-x} catalysts with varying Pt loadings: 0.2%, 0.4%, 1.0% and 2.0%. (a) XRD diffraction patterns. (b-c) N₂ adsorption-desorption isotherms, and (c) pore size distribution profiles for the four catalysts. (d1-d3) AC-HAADF-STEM images of 0.2% Pt/ α -MoC_{1-x} catalyst. (d4) Line-profile intensities of Pt atomic layers corresponding to panel (d1). (e1-e3) AC-HAADF-STEM image of 1.0% Pt/ α -MoC_{1-x} catalyst. (e4) Line-profile intensities of Pt atomic layers corresponding to panel (e3). Pt₁ species are highlighted with green dashed circles, and Pt clusters (Pt_n) are highlighted with red ellipses.

(11) Are no Pt atoms or clusters visible in the STEM imaging? Surely some would be visible based on the modeled size of these in subsequent parts?

Response:

We are very sorry that the unclear introduction to the processing and analysis of AC-HAADF-STEM characterization data has caused confusion for readers. Therefore, the following modifications were made in the manuscript.

On the 0.2% Pt/ α -MoC_{1-x} catalyst, Pt primarily exists as single atoms and small clusters (Fig. 1(d1-d3)). The line profile intensity of the isolated atoms (Fig. 1(d4), Fig. 1(e4), and Figs. S7-S8) corresponds to Pt atom sizes, indicating that the small clusters form an atomic monolayer on the α -MoC_{1-x} surface, rather than being embedded within the α -MoC_{1-x} structure (details in Supplementary Note 1). In the 1.0% Pt/ α -MoC_{1-x} catalyst, Pt also shows a monolayer distribution, resembling a “bamboo raft”

arrangement, but with significantly larger clusters than in the 0.2% Pt/ α -MoC_{1-x} sample.

(Manuscript, Page 7, line 14-21)

Fig. 1 Textural properties of the Pt/ α -MoC_{1-x} catalysts with varying Pt loadings: 0.2%, 0.4%, 1.0% and 2.0%. (a) XRD diffraction patterns. (b-c) N₂ adsorption-desorption isotherms, and (c) pore size distribution profiles for the four catalysts. (d1-d3) AC-HAADF-STEM images of 0.2% Pt/ α -MoC_{1-x} catalyst. (d4) Line-profile intensities of Pt atomic layers corresponding to panel (d1). (e1-e3) AC-HAADF-STEM image of 1.0% Pt/ α -MoC_{1-x} catalyst. (e4) Line-profile intensities of Pt atomic layers corresponding to panel (e3). Pt₁ species are highlighted with green dashed circles, and Pt clusters (Pt_n) are highlighted with red ellipses.

Fig. S7 The line-profile intensities of Pt atomic layers of 0.2% Pt/ α -MoC_{1-x} sample.

Fig. S8 The line-profile intensities of Pt atomic layers of 1.0% Pt/ α -MoC_{1-x} sample.

(12) The bamboo raft and presence of single layers of Pt are described, but where are they in the images?

Response:

We are very sorry that the description of Fig. 1 was not clear. The bamboo raft and presence of single layers of Pt atoms are shown in the red circles in Fig. 1. As the loading increases, the size of bamboo raft-like monolayer Pt clusters gradually increases.

(13) Why were Mo⁶⁺ and Mo⁴⁺ peaks omitted from the XPS fit. There is significant overlap between carbidic peaks and high oxidation state peaks associated with Mo-O as is widely reported in the literature that are very close to the 2 peaks that were fit. These peaks would be expected, particularly if the sample was passivated in 1% O₂ as the penetration depth of the XPS is limited.

Response:

We fully agree with the reviewer that there is an overlap between Mo carbide and Mo oxides when the sample is passivated. We are fully aware of the challenges in studying Mo carbides using XPS. To address these challenges and obtain intrinsic surface properties of Mo carbides, we carefully prepare the samples under anaerobic conditions. Firstly, the m% Pt/ α -MoC_{1-x} catalysts were prepared in a fixed-bed reactor and then transferred to a glove box filled with argon gas. The samples were

subsequently moved from the glove box to the XPS instrument using an in-situ transfer vessel. By avoiding surface oxidation, no Mo^{6+} and Mo^{4+} peaks were detected. Literature studies have also demonstrated the absence of Mo^{6+} and Mo^{4+} peaks in unoxidized samples (Fig. R5) 2, 3, 5.

Fig. R5 XPS characterization results, (a) NAP-XPS results of the 2 wt% (Pt₁-Pt_n)/α-MoC catalyst before activation (black, acquired in ultrahigh vacuum), after activation (red), under 0.5 mbar H₂O at 303 K after 5 mbar water treatment at 303 K for 3 h (blue), and under 0.5 mbar CO at 303 K after 5 mbar CO treatment of a water-treated sample at 303 K (green), 323 K (purple), 373 K (brown) and 423 K (cyan) for 10 min²; (b) In situ XPS characterization of the 2% Ni/α-MoC catalyst under methanol and water reaction condition³; (c) XPS spectra of the catalysts⁵.

(14) Why were DRIFTS experiments not conducted to evaluate the difference between Pt and Mo-based CO binding? This would be a fairly straightforward way to determine differences in binding sites no?

Response:

The DRIFTS experiment is a good method for characterizing CO adsorption properties. However, the Pt/α-MoC_{1-x} material appears charcoal black in colour (as shown in Fig. R6), and there are no other characteristic peaks except for the CO gas phase peak due to its high absorbance (as shown in Fig. R7(a)). In addition, we attempted to adjust the colour by adding KBr during the testing process, but still failed

(as shown in Fig. R7(b)). Therefore, we regret to abandon the DRIFTS experiments to determine the adsorption sites of CO molecules on the catalyst surface.

Fig. R6 Pt/ α -MoC_{1-x} sample

Fig. R7 DRIFTS studies of CO bonds on Pt/ α -MoC_{1-x} under CO gas conditions, (a) KBr not added; (b) Add KBr

(15) What is “the law of CO adsorption”?

Response:

We apologize for the issue of using multiple words to describe the same meaning in the manuscript. “the law of CO adsorption” and “CO adsorption rule” all refer to a rule followed by the adsorption sites and configurations of CO molecules on Pt/ α -MoC_{1-x} surfaces, and the adsorption rules will change with the type of catalyst. The entire manuscript has been uniformly revised and the “CO adsorption rule” was chosen. Its definition was added in the manuscript (Manuscript, Page 16, line 19-21).

Reference (Corresponds to the number marked in yellow in the text)

1 Lin L, Wu Z, Rui G, Yao S, Ding M. Low-Temperature hydrogen production from water and methanol using Pt/ α -MoC catalysts. *Nature* **544**, 80-83 (2017).

2 Zhang X, *et al.* A stable low-temperature H₂-production catalyst by crowding Pt on α -MoC. *Nature* **589**, 396-401 (2021).

3 Lin L, *et al.* Atomically Dispersed Ni/ α -MoC catalyst for hydrogen production from methanol/water. *J. Am. Chem. Soc.* **143**, 309-317 (2021).

4 Liang P, *et al.* Simple synthesis of ultrasmall β -Mo₂C and α -MoC_{1-x} nanoparticles and new insights into their catalytic mechanisms for dry reforming of methane. *Catal. Sci. Technol.* **7**, 3312-3324 (2017).

5 Cai F, *et al.* Low-temperature hydrogen production from methanol steam reforming on Zn-modified Pt/MoC catalysts. *Appl. Catal. B Environ.* **264**, 118500 (2020).

6 Sun X, *et al.* In Situ Investigations on Structural Evolutions during the Facile Synthesis of Cubic α -MoC_{1-x} Catalysts. *J. Am. Chem. Soc.* **144**, 22589-22598 (2022).

Responses to Reviewers' Comments

Dear Reviewers,

We are very happy that the article was accepted by nature communications. Thank you for your valuable suggestions on the article during the publication of this article, so that the article can be improved and finally accepted.

Reviewer #2:

The response of the authors to my previous comments is satisfactory. They have substantially improved the quality of the article and, therefore, I recommend acceptance for publication.

We appreciate the positive comments from the reviewer. Again, thanks very much for the reviewer's careful review and valuable suggestions.

Reviewer #3:

The authors have suitably addressed my suggestions. If this is also the case for the other reviewers, whose comments are outside of my area of expertise, I am comfortable with publication.

Again, thank you very much for your comments and professional advice. These opinions help to improve academic rigor of our article.